# A comparison of estimates of global carbon dioxide emissions from fossil carbon sources

Robbie M. Andrew[1]

[1]CICERO Center for International Climate Research, Oslo 0349, Norway

*Correspondence to*: Robbie M. Andrew (robbie.andrew@cicero.oslo.no)

**Abstract.** Since the first estimate of global $CO_2$ emissions was published in 1894, important progress has been made in the development of estimation methods while the number of available datasets has grown. The existence of parallel efforts should lead to improved accuracy and understanding of emissions estimates, but there remains significant deviation between estimates and relatively poor understanding of the reasons for this. Here I describe

the most important global emissions datasets available today and – by way of global, large-emitter, and case examples – quantitatively compare their estimates, exploring the reasons for differences. In many cases differences in emissions come down to differences in system boundaries: which emissions sources are included and which are omitted. With minimal work in harmonising these system boundaries across datasets, the range of estimates of global emissions drops to 5%, and further work on harmonisation would likely result in an even

lower range, without changing the data. Some potential errors were found, and some discrepancies remain unexplained, but it is shown to be inappropriate to conclude that uncertainty in emissions is high simply because estimates exhibit a wide range. While 'true' emissions cannot be known, by comparing different datasets methodically, differences that result from system boundaries and allocation approaches can be highlighted and set aside to enable identification of true differences, and potential errors. This must be an important way forward

in improving global datasets of $CO_2$ emissions. Data used to generate figures 4–19 are available at https://doi.org/10.5281/zenodo.3687042 (Andrew, 2020).

## 1    Introduction

Since the first known estimate of global anthropogenic emissions of $CO_2$ was made in the early 1890s, methods have substantially improved, detail has increased, and additional emissions sources have been included.

Meanwhile, with international agreements to mitigate climate change, the production of such estimates has grown beyond the realm scientific enquiry to become a critical input to policy.

Fossil $CO_2$ emissions occur when fossil carbon compounds are broken down via combustion or other oxidation processes. Most of these fossil compounds are in the form of fossil fuels, such as coal, oil, and natural gas. In



addition are fossil carbonates, such as calcium carbonate and magnesium carbonate, which are used as feedstocks in several important industrial processes – including cement production – and whose decomposition also leads to emissions of $CO_2$.

Every year several global emissions datasets are updated and present different estimates of $CO_2$ at both national and global levels, but there is little information available about why these estimates differ, and sometimes the range between these estimates is merely assumed to represent uncertainty, suggesting that no more can be known. In fact, there are core reasons why the estimates from these different datasets differ, but these are largely buried in the documentation and have not previously been comprehensively analysed.

The accumulated global emissions of carbon dioxide are drawing precariously close to the best estimates of the total budget available before the world crosses certain temperatures agreed to in international negotiations (CONSTRAIN, 2019). A detailed understanding of these different annual estimates – to unravel the 'range is uncertainty' tangle – is therefore important in efforts to reduce uncertainty and improve our understanding of the requirements before global society.

There are identifiable reasons why estimates differ between datasets, but this requires a close examination of the datasets' documentation and quantitative comparison of the data themselves to determine the significance of differences in sources and methods used. In particular, not all datasets attempt to be comprehensive either geographically or by including all emissions sources.

In this article I will discuss datasets whose spatial resolution is at the country level, but there exist several datasets that are further disaggregated to grids of varying resolution. These include CDIAC (Andres et al., 1997;Andres et al., 2016a), ODIAC (Oda and Maksyutov, 2011;Oda et al., 2018) EDGAR (Janssens-Maenhout et al., 2012), FFDAS (Asefi-Najafabady et al., 2014), CEDS (Hoesly et al., 2018), and PKU-FUEL (Chen et al., 2016). Many of these gridded datasets use existing country-level datasets as primary input data and will therefore have similar attributes to the datasets discussed in this article. An overview of some gridded dataset was presented by Andres et al. (2012), and recent assessments of uncertainty in gridded emissions are presented by Andres et al. (2016b) and Oda et al. (2019).

Because fossil fuel $CO_2$ emissions are largely connected with energy, which is a closely tracked commodity group with its critical role in economic activity, there is a wealth of underlying data that can be used for estimating emissions. However, differences in collection, treatment, interpretation, inclusion, and various factors such as carbon contents and fractions of oxidised carbon, can lead to significant differences in estimates of emissions between datasets.



Several comparisons have been performed before, notably Marland et al. (1999), who compared CDIAC and EDGAR, Andres et al. (2012), who made a high-level comparison of several datasets, and Ciais et al. (2010), who compared datasets for the EU. The most complete comparison of global datasets to date is that of (Macknick, 2011), who made quantitative comparisons of CDIAC, EDGAR, IEA, BP, and EIA, including

discussion of the differences in underlying energy datasets. This study is now almost 10 years old, and methodologies of each of those datasets have changed in the intervening time, while new datasets are also available.

This paper summarises early efforts to quantify global CO2 emissions, before moving to current datasets, starting by discussing differences that can be expected *a priori*, describing the important IPCC inventory

guidelines, summarising the publicly available emissions datasets, and then comparing these in some detail and explaining quantitative differences where possible. Because many of these products have appeared and will continue to appear in Earth System Science Data, readers can apply this review as a detailed guide to most recent emissions compilations and products as published in this journal.

## 2    Early estimates

In the earliest years the carbon cycle of interest was the long-term balancing of geological, extra-terrestrial, and natural-system inputs and outputs, primarily to understand the mysteries of the ice ages. While geological interest remains, interest in the human perturbation of the global carbon cycle has grown as anthropogenic emissions – once considered negligible – have grown to a very significant degree. Accurately gauging the magnitude of this perturbation is of key importance in understanding how current and future climatic changes

relate to our historical and current emissions. Over more than a century, the quality of estimates of global $CO_2$ emissions have steadily improved as greater understanding and greater effort have been brought to bear on the problem.

In the early 1890s, Swedish geochemist Arvid Högbom was possibly the first to consider the global geochemical carbon cycle, and presented some of his thoughts to the Swedish Chemical Society, later published in the

Swedish Chemistry Journal (Högbom, 1894). Högbom briefly considered whether combustion of fossil fuels might perturb the carbon cycle, estimating that emissions at that time were 0.5 GtC (see Supplementary Information for details), and determining that this was insufficient to have any effect on atmospheric concentrations because it would merely compensate for $CO_2$ consumed in the continuous formation of carbonates.

Svante Arrhenius, inspired by Högbom's lecture, initially accepted his main conclusion that the short-term carbon cycle was in balance, but considered what might happen if fossil emissions were to further increase



(Arrhenius, 1908). In so doing he presents of emissions estimates from the global combustion of coal of 0.51 GtC in 1890, 0.55 GtC in 1899, and 0.89 GtC in 1904.

Guy Callendar, investigating the influence of fossil $CO_2$ on temperature, stated without support that annual emissions of $CO_2$ at the time amounted to 4.3 $GtCO_2$, and that the cumulative emissions over the previous fifty

years were 150 $GtCO_2$ (Callendar, 1938).

Gilbert Plass stated, again without support, that the combustion of fossil fuels at that time were adding 6 $GtCO_2$ annually (Plass, 1956). He also listed other human activities that release carbon, unfortunately without any quantification: "the clearance of forests, the drainage and cultivation of lands, and industrial processes such as lime burning and fermentation" (Plass, 1956, p.379).

In 1957, Roger Revelle and Hans Suess, interested in the fate of the carbon dioxide added to the atmosphere by human activities, estimated emissions from fossil-fuel combustion per decade, from the 1860s through 1940s (Revelle and Suess, 1957). The methods were not given, but reference was elsewhere given to a recent United Nations report 'World requirements of energy, 1975–2000' presented at the International Conference on Peaceful Uses of Atomic Energy, held in Geneva in 1955 (United Nations, 1956), and this is most likely the

source of the energy data.

Importantly, all the foregoing estimates appear to have implicitly assumed in their calculations that fossil fuels were composed almost entirely of carbon. It wasn't until the 1965 report by the President's Science Advisory Committee Panel on Environmental Pollution that sources and carbon contents by fuel type are provided (Revelle et al., 1965). They also broke emissions down by main fuel category (coal, oil, gas). In addition,

specific sources of energy data are clearly stated.

Baxter and Walton (1970), looking to explain the decline in the fraction of isotope [14]C in the atmosphere, presented annual fossil carbon emissions, including, for the first time, from cement production. While methods and sources are reasonably clearly presented, some errors in interpretation of sources led to highly inflated estimates of emissions from lignite.

Broecker et al. (1971), looking at uptake of $CO_2$ by the oceans, present decadal global emissions by coal, oil, and gas, and were the first to make explicit their assumption of complete oxidation of all fossil fuels.

Keeling (1973) was dissatisfied with the lack of rigour in previous emissions assessments, and substantially increased the detail of analysis. He identified that earlier studies had greatly overestimated emissions from coal and lignite by using inflated carbon contents, introduced adjustments for non-energy uses of fuels and losses, and

performed an uncertainty assessment. Keeling's methods shaped the methods for estimating $CO_2$ emissions for the following decades.



Rotty (1973) introduced estimates for flaring and venting of natural gas and $CO_2$, although because of data limitations venting of natural gas could not be separated out and the methane content was therefore assumed to be oxidised immediately to $CO_2$, an assumption that continued in later datasets. Rotty also pointed to an alternative source of energy data, demonstrating that the two sources were in agreement when close attention

was paid to definitions. Later, Rotty (1983) presented the first estimates of sub-global $CO_2$ emissions using apparent consumption of energy – production adjusted for international trade – rather than the energy production data used in all previous studies.

Marland and Rotty (1984) re-examined the method of Keeling (1973) and produced slightly revised parameters, still constant in time, an assumption that appeared valid given the observational data available at the time.

Importantly, they were the first to make use of new energy data from the United Nations that was already in units of energy, having been converted from physical units using country-specific factors. This avoided the use of global-average conversion factors that previous estimates had been based on. The method developed by Marland and Rotty (1984) has been used – with some modifications and improvements – by CDIAC right through until its 2019 release.

**3    Potential reasons for differences between datasets**

Before turning to an exploration of the actual differences between emissions datasets, I first discuss some of the reasons for which datasets should be expected *a priori* to differ based on what is already known. Many of these hinge on what can be called 'system boundaries.'

The term 'system boundary' is found in the life-cycle assessment (LCA) literature where it describes the scope

of analysis: which activities in a process or supply chain are included and which are omitted (Baumann and Tillman, 2004). Here I use the term to describe the categories of emissions that are included in each dataset, and the way in which they are distinguished when presented in more detail. There are many aspects to these, which I will discuss in turn, limiting myself to $CO_2$ emissions data.

There are three main physical sources of anthropogenic carbon dioxide: oxidation of fossil fuels, land-use

change (e.g., deforestation), and decomposition of (fossil) carbonates (Friedlingstein et al., 2019). All emissions datasets include fossil fuels, while fewer include either land-use change or carbonates.

Decomposition of carbonates occurs in production of cement, lime, and glass, but also in steel manufacture where carbonates are used as a flux agent to facilitate removal of impurities, and in flue gas desulphurisation (REF). Datasets may exclude carbonate emissions entirely, include emissions only from cement production (e.g.,

CDIAC), or from all carbonate decomposition (e.g., EDGAR).



Several emissions datasets are relatively simple extensions of energy datasets (e.g., IEA, EIA, BP), and their primary purpose is to show the emissions associated with consumption of energy, rather than to provide a comprehensive picture of all emissions of $CO_2$. Most emissions of $CO_2$ are from fossil fuels, with emissions from land-use change and carbonates combined currently amounting to about 13% and 5% of the global total, respectively (Friedlingstein et al., 2019;Crippa et al., 2019).

While most datasets focus on combustion of fossil fuels, some extend the definition to all oxidation of fossil fuels. While combustion is one form of oxidation, other forms exist, such as in chemical processes where hydrocarbons are used as a source of carbon or as a reducing agent. This distinction generally hinges on whether the fossil fuel is primarily required as an energy source (energy released by combustion) or as an agent in a chemical process.

Particularly coal consumption in the metals industry can be considered as either combustion or as a reducing agent. Coke used in refining iron ore is critical as a reducing agent, but also serves as an energy source. However, when coal and oil are used to make carbon anodes for aluminium smelting, the oxidation of the anode that occurs during smelting is not considered combustion.

The entities to which emissions are assigned vary between datasets. For example, some parties have different geographic and economic extents under the Kyoto Protocol and the UNFCCC, and therefore submit more than one inventory to the UNFCCC. These include Denmark, France, and the United Kingdom (EEA, 2019). These make differences of less than 2% for individual countries. The European Union also submits two sets of inventories: EUA (Convention: strictly EU territory) and EUC (Kyoto Protocol: includes also Iceland and overseas territories of member states). Similarly, the United States reports include Puerto Rico and other territories when submitted to the UNFCCC.

Moreover, emissions can be limited either to geographical areas or economic activities. Inventories, as for example submitted to the UNFCCC, cover geographical areas (akin to Energy Balances), while Accounts cover economic activities (akin to Energy Accounts) (UNSD, 2018). Accounts must be adjusted for the activities of foreign nationals and companies within the territory (e.g. emissions from tourists driving cars), and activities of nationals and national companies in other territories. Accounts follow the definitions of the System of National Accounts, used, among other things, for calculating GDP (European Commission et al., 2009).

With regard to these country definitions, the allocation of emissions from combustion of international bunker fuels has been particularly problematic. While energy data are collated as to which country sells bunker fuels, this is very poorly related to which country has responsibility for the combustion of those fuels. Various methods have been proposed to allocate these emissions, such as to the country whose flag a ship operates under, or that





which the owner of the ship is a tax resident in, or those that operate the ship, of even those who purchase the goods borne by the ship (Heitmann and Khalilian, 2011). However, none of these is clearly superior to the others, and they can result in very different distributions of these emissions. This is in effect why the international aviation and maritime industries have been largely excluded from negotiations and are acting partly

independently on a global basis (UNFCCC, n.d.). See Supplementary Information for further details on the inclusion of bunker fuels.

Further methods of allocating emissions have been devised, such as re-allocating through economic supply chains to the point of final consumption, so-called consumption-based emissions, and variants (e.g., Davis and Caldeira, 2010;Andrew et al., 2013). These alternatives have not yet obtained international acceptance, although

official reporting at national level does occur in, e.g., UK (DEFRA, 2019), Sweden (Björk et al., 2018), and France (SOeS, 2012;I4CE, 2018).

The ways in which national or global emissions are presented in more detailed form can vary substantially between datasets. While the IPCC Guidelines set a clear method for differentiating between "sectors of [the] economy" (Penman et al., 2006, p. 4), these sectors are quite different to those understood by economists. The

Energy sector, for example, includes most combustion of energy, whether the activities are undertaken by enterprises whose main activity is energy production or not. All household combustion of gasoline in private transportation is included in the Energy sector, whereas under economic accounts such activities would be accounted to the Household sector. Agricultural emissions, under the IPCC methodology, do not include such activities as driving tractors or heating glasshouses. So, while in economics a 'sector' is a grouping of similar

economic actors, in the IPCC Guidelines a 'sector' is a grouping of activities. Some other datasets do assign emissions to economic activities. Further, breakdown by type of fossil fuel can vary, with the use of Solid, Liquid, and Gaseous fuel categories as distinct from Coal, Oil, and Natural Gas (see section 5.4).

The time period over which emissions are accounted can vary. While all modern datasets present annual emissions, some also report sub-annual periods. More importantly, while most countries' data are reported for

calendar years (from 1 January to 31 December), some are reported for financial years. In the IEA's data, which is probably representative of most datasets because of non-independent original sources, non-calendar year data are reported for Bangladesh, Egypt, India, Nepal. For India, by far the most significant of these, IEA's data for 2016 represents the financial year 1 April 2016 – 31 March 2017 (the majority of this period falling in 2016), which would be called the 2017 year in India (the financial year ending in 2017, FY17).

One final category of system boundaries is the inclusion of confidential data. At detailed levels some countries may withhold and aggregate reporting of emissions from certain activities for strategic reasons. While these are





generally included at aggregate level, emissions from military activities are known sometimes to be withheld entirely. The IEA "has found that in practice most countries consider information on military consumption as confidential and therefore either combine it with other information or do not include it at all" (IEA, 2018c, p. 55). While some confidential emissions might be excluded from national accounts, energy production statistics

most likely cover all energy produced so that estimates of global fossil $CO_2$ emissions would not be expected to exclude military energy consumption.

Different methods can be used to estimate emissions, based on different original data sources. The most important distinction is between the Sectoral Approach and the Reference Approach, applying only to emissions from fossil fuels. While the Sectoral approach is based on detailed demand-side energy data (a bottom-up

calculation, starting with as much detail as possible, typically sales data), the Reference approach is based on much less detailed supply-side data (a top-down calculation, typically using national production, international trade, and stock change data). At the national level, estimates generated by the Sectoral Approach are definitive, while those under the Reference Approach are used as a partially independent cross check (Treanton et al., 2006b).

**4    IPCC Inventory Guidelines**

The Intergovernmental Panel on Climate Change (IPCC) provides comprehensive guidance for compiling emissions inventories for all sources of emissions, which must be used in reporting to the UNFCCC (UNFCCC, 2019b).

The Guidelines are built on decades of efforts and expertise in compiling emissions estimates and are designed to

be flexible to suit countries' specific needs. Work began on The Guidelines in 1991 by Working Group 1 of the IPCC under the IPCC/OECD/IEA Programme on National Greenhouse Gas Inventories, with the first edition approved in 1994 and adopted the following year. A revision to these was published in 1996 (Houghton et al., 1996), and a new edition published in 2006 (Eggleston et al., 2006), with later amendments (e.g., the wetlands supplement, Hiraishi et al., 2013). In 2019 a 'refinement' to the guidelines was released, although with little

material consequence for fossil emissions beyond some changes to fugitive emissions calculations and clarifications on biofuel emissions (IPCC, 2019).

The methodology is divided into three 'Tiers', where Tier 1 uses supplied default emission factors applied to national activity data, Tier 2 uses national emission factors, and Tier 3 uses national models and/or direct measurements. The Tier 1 approach is often used by compilers of international inventories because they can do

so using existing international datasets of activity, such as energy or agriculture databases.



The IPCC Guidelines divide emissions into 'sectors', the most important of which for $CO_2$ emissions are the Energy sector and the Industrial Processes and Product Use (IPPU) sector. The Energy sector includes emissions from activities in the energy industry (e.g., electricity production, flaring on oil platforms) and all uses of energy sources primarily for energy purposes. The IPPU sector, in contrast, includes both emissions from decomposition

5    of carbonates (e.g., cement production) and non-energy uses of fossil fuels (e.g., carbon anodes in aluminium smelting). Smaller amounts of $CO_2$ are also reported in the Agriculture sector, as emissions from use of urea and lime.

While the 1996 Guidelines included a default fraction of carbon stored (sequestered) from non-energy use (allowing for some to be oxidised at some point), the 2006 Guidelines removed these, effectively setting the

10    fraction stored to 1.0 for all products. This was because "in most instances, emission inventory compilers had no "real" information as to whether this correction was actually applicable" (IEA, 2018a, p. I.22).

## 5    Detailed descriptions of emissions data sources

Figure 1 shows the main global energy datasets, primary global emissions datasets, and secondary (derived from primary) emissions datasets. The most important data source type for emissions estimates is energy data, which

15    are ultimately derived from heterogeneous national sources. The IEA and Eurostat have developed questionnaires that are sent to at least 61 countries: all members of the Organisation for Economic Co-operation and Development (OECD), European Union (EU), United Nations Economic Commission for Europe (UNECE), "and a few others". These identically completed questionnaires are returned to the IEA, the UN, and (for certain countries) Eurostat (IEA, *no date-a*).

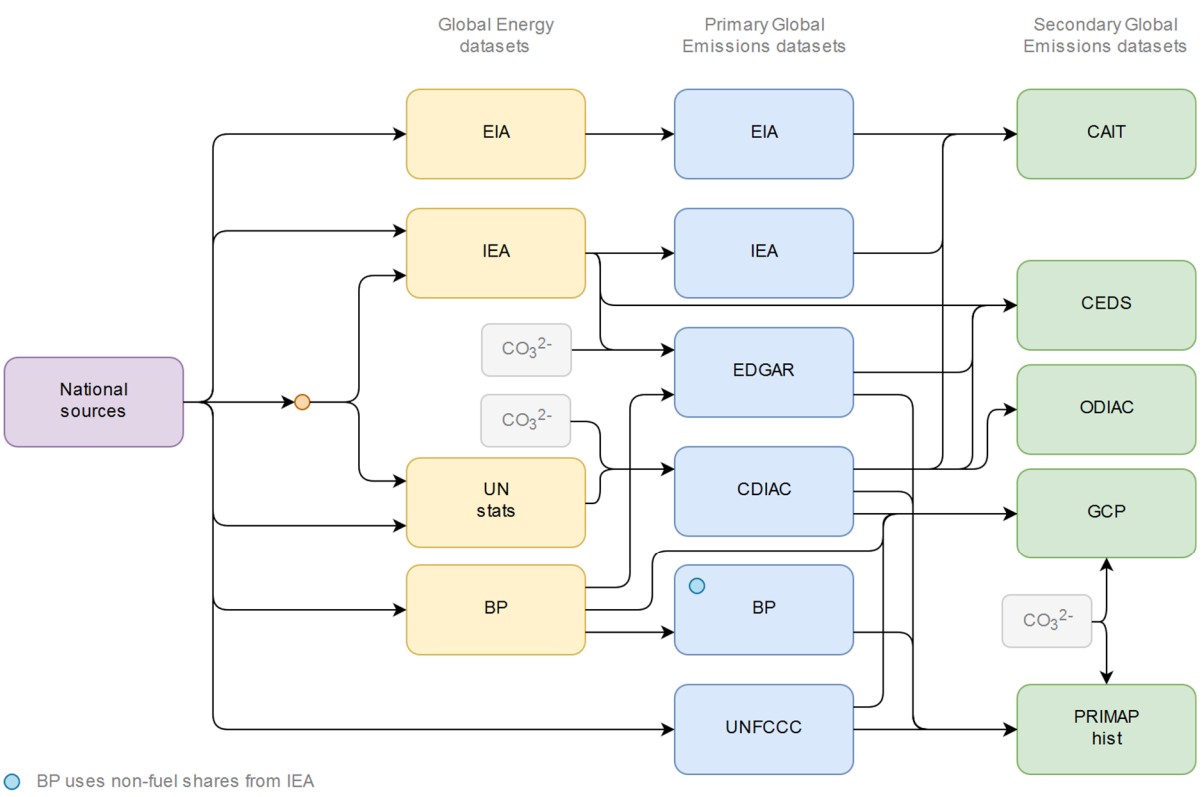

**Figure 1: Dependencies of selected global energy and CO₂ emissions datasets. Here a 'primary' emissions dataset is one that calculated emissions directly from energy data, rather than collating emissions estimates from other sources. In addition to energy data sources, some emissions datasets include emissions from carbonates, which rely on other data sources. Some national data are first collated by regional organisations.**

In the following sections I will describe each of the main global emissions datasets in turn. Table 1 summarises the datasets and the versions analysed in detail.





**Table 1: Summary of datasets included in this analysis.**

| Dataset | Short name | Version | Reference | DOI / URL |
|---|---|---|---|---|
| Global Carbon Project | GCP | 2019 | Friedlingstein et al., 2019 | 10.18160/GCP-2019 |
| Energy Information Administration International Energy Statistics | EIA | 2020 | EIA, 2020 | https://www.eia.gov/international/data/world |
| International Energy Agency $CO_2$ Emissions from Fuel Combustion | IEA | 2019 | IEA, 2019b | https://webstore.iea.org/co2-emissions-from-fuel-combustion-2019-highlights |
| BP Statistical Review of World Energy | BP | 2019 | BP, 2019 | https://www.bp.com/en/global/corporate/energy-economics/statistical-review-of-world-energy.html |
| Community Emissions Data System | CEDS | v_2019_12_23 | Hoesly et al., 2018 | 10.5281/zenodo.3606753 |
| Emissions Database for Global Atmospheric Research | EDGAR | v5.0 | Crippa et al. 2019 | https://edgar.jrc.ec.europa.eu/overview.php?v=50_GHG |
| Annex 1 Common Reporting Format | CRF | 20-Apr-19 | UNFCCC, 2019a | https://unfccc.int/process-and-meetings/transparency-and-reporting/reporting-and-review-under-the-convention/greenhouse-gas-inventories-annex-i-parties/national-inventory-submissions-2019 |
| CDIAC Global, Regional, and National Fossil-Fuel $CO_2$ Emissions | CDIAC | 2019 | Gilfillan et al., 2019 | https://energy.appstate.edu/research/work-areas/cdiac-appstate |
| Potsdam Real-time Integrated Model for probabilistic Assessment of emissions Paths | PRIMAP | 2.1 | Gütschow et al., 2019b | 10.5880/PIK.2019.018 |



### 5.1 BP's Statistical Review of World Energy

BP produced its first limited-circulation Statistical Review of World Energy in 1952 (BP, 2011). In recent years the BP Review has been highly anticipated primarily because it is the earliest data release to cover global energy and fossil-fuel $CO_2$ emissions, published in June of each year with data up to the previous year. The dataset is widely used, something that is facilitated by its being freely available, and its publication in full in Microsoft Excel format. Since 2007 the Review has been produced in collaboration with the Centre for Energy Economics Research and Policy (CEERP) at Heriot-Watt University,

Scotland (Heriot Watt University, 2017).

Energy data are sourced directly from countries, although there is little documentation of specific sources. In the most recent edition, data were reported for 80 separate countries in addition to further regional groupings, from 1965 to 2017 (BP, 2018). It appears that emissions were first included in the second release of the 2009 edition. Prior to the 2016 edition, emissions of $CO_2$ were calculated simply using a single emission factor for each of oil, gas, and coal, taking no account of consumption

for non-combustion purposes (e.g., bitumen). From 2016 this has been revised to use the default emission factors for each product type from the IPCC 2006 Guidelines, with biofuels assumed to be carbon neutral. In addition, non-combusted energy is now removed using shares from IEA's *Energy Balances* (BP, 2017). The main consequence of this change is a decrease in emissions from oil in particular, because of high non-fuel use in oil.

BP provides national total emissions, without a breakdown by fuel type or sector. International bunker fuels are not
separately reported but included in national emissions. Emissions from venting and flaring are excluded, while own use (e.g., on oil rigs) is included (pers. comm., CEERP, February 2019).

In house, BP have energy data at a more disaggregated level than those reported (coal, oil, and natural gas), and emissions are calculated from these more disaggregated data. While oil is fully disaggregated, Coal is divided into Hard Coal and Brown Coal (coke is assigned to hard coal), and Gas is just Natural Gas (pers. comm., CEERP, February 2019). This means

that both the non-fuel-use (NFU) shares and the emission factors are applied at these disaggregated levels. A consequence of this is that emissions from Oil and Natural Gas should lie close to IEA's estimates, with differences deriving from differences in source energy data, while Coal might deviate because of a lower level of disaggregation. The NFU share from the most recent IEA data year are used in cases where BP data extend beyond the period of the IEA data (pers. comm., CEERP, February 2019).

Because BP's dataset is the first to come out with the previous year's energy data, growth rates derived from its data can be used to extend emissions datasets (Myhre et al., 2009), and this is done by CDIAC, GCP, EDGAR, and PRIMAP-Hist.

BP's global emissions estimate is calculated as the sum of all country estimates.

BP provides no quantitative assessment of uncertainty associated with its emissions dataset.

### 5.2 Climate Analysis Indicators Tool (CAIT)

The World Resources Institute (WRI) developed CAIT, collating data from other emissions datasets. The 2015 edition included 185 countries (WRI, 2015), with emissions by sector for 1990–2014, and country total emissions for 1850–2014.





For CO$_2$, IEA's sectoral approach emissions estimates are used directly for the 135 countries covered by that dataset, starting from 1971. CDIAC is used from 1850 to 1970 for all countries — with estimates prior to 1850 deemed to have insufficient geographic coverage — and from 1971 to 2011/12 for countries not present in IEA's dataset. Because Lesotho's data in

CDIAC begin only in 1990, CAIT uses EIA for Lesotho for 1980 to 2012. EIA data are also used for 2012 for all countries for which 2012 was not present in the other two datasets, potentially introducing discontinuities. EIA data are also used for emissions from flaring. UNFCCC inventories are not used in the main dataset because of their limited geographic coverage but are presented separately.

### 5.3    Community Emissions Data System (CEDS)

The Community Emissions Data System (Hoesly et al., 2018) is intended to be an open-source emissions data production system, although the full system requires access to IEA's energy data. It produces annual national, sectoral, and monthly gridded emissions of a number of greenhouse gases and pollutants, including CO$_2$. CEDS estimates are to be used in the sixth round of Coupled Model Intercomparison Project for climate models (CMIP6), and were initially limited to anthropogenic aerosol and aerosol and ozone precursor compounds (Smith et al., 2015), but have since expanded.

Emissions of CO$_2$ are primarily derived from IEA's World Energy Statistics in physical units, using emission factors from CDIAC and EIA. Cement emissions are taken directly from CDIAC. Estimates for countries for which official (or near official) estimates are available are scaled to those official estimates during the periods they are available.; this scaling maintains the proportions of any greater sectoral disaggregation available in the IEA energy data.

Emission factors from CDIAC are applied for coal and natural gas combustion, from Boden et al. (1995) (and therefore also

Marland and Rotty (1984);Marland and Boden (1993)). For China a lower coal oxidation factor was used, based on specific research there (Liu et al., 2015). For liquid fuels (heavy, medium, and light oils) and coal coke, emission factors are taken from the EIA. Emission factors are modified by fuel-specific fractions oxidized, following CDIAC's documented methodology.

CEDS uses an independent estimate of international marine bunker fuel emissions, combining estimates from several sources

that have used bottom-up methods based on ship activity and fuel consumption rather than on reported sales of bunker fuels. CEDS' estimates of these emissions in recent years is more than 50% higher than those reported by the IEA in some years (Figure SI-28). There is however the potential for double counting of emissions here based on the possibly incorrect assumption that underestimated emissions from international marine bunkers means that those emissions are omitted in other sources, rather than that they are misallocated. See Supplementary Information for more discussion of bunker fuels.

Emissions calculated using IEA energy data are 'default emissions' for 1960/71–2014. These are then scaled to EDGAR, then to 'national inventories' where available, then extrapolated historically using CDIAC – with some minor corrections to CDIAC's data – and proxy activity data. For China, the emissions dataset MEIC (Li et al., 2017) is considered a 'national inventory' and China's emissions are scaled to MEIC for the years 2008, 2010, and 2012.

CEDS reports a breakdown of emissions by fuel type at the global level, but not at national level, due to rights restrictions

tied to the energy data from the IEA.



CEDS is one of two global $CO_2$ datasets (the other being EDGAR) that include global estimates of emissions from all carbonate decomposition, not just cement production, and thereby covers all fossil carbon sources.

CEDS' global emissions estimate is calculated as the sum of all country estimates. The period covered by the dataset is 1750–2014.

Hoesly et al. (2018) discuss uncertainty at some length, but quantitative estimates of uncertainty have been completed yet.

### 5.4 CDIAC

The emissions dataset of the Carbon Dioxide Information Analysis Center (CDIAC) at Oak Ridge National Laboratory has been widely used, and some aspects of its construction methodology were incorporated into the Tier 1 approach in the first IPCC Guidelines (Haukås et al., 1997). The IPCC's Fifth Assessment Report used CDIAC's emissions estimates when
reporting both long-term and short-term emissions trends (Ciais et al., 2013).

The dataset has a long heritage (see Section 2), and its long pedigree and long time-series with a consistent methodology is probably one of the core reasons the CDIAC dataset remains widely used.

The CDIAC dataset has been updated annually, with the most recent release in 2019 with data for 1751–2016. In 2016 it was announced that the US Department of Defense would be withdrawing funding for CDIAC, throwing the dataset's future into
doubt, but it has since been taken up again by Appalachian State University (ASU). The 2018 and 2019 releases were made available on the ASU website (Boden et al., 2018;Gilfillan et al., 2019), and plans are in place to continue regular updates in future (pers. comm., Gregg Marland, April 2019).

CDIAC's estimates are primarily derived from UN energy data, which in more recent years were in most cases identical to IEA data, except for UN's addition of data for a number of small countries (see Figure 1). The emissions data include
estimates in five categories: Solid, Liquid, Gas, Cement, and Flaring, with cement emissions derived from USGS cement production data.

Separate methods are used to derive global and national estimates, and these methods have evolved since the dataset's documenting articles. For global estimates, CDIAC uses energy production data rather than consumption data based on the assumption that consumption data are more uncertain because they rely on more uncertain international trade data. Through
the 2018 edition, global energy production was not adjusted for stock changes, but in the 2019 edition an adjustment for global stock changes was introduced for historical data back to 1992 in light of very high coal stock changes in 2016 (pers. comm., Gregg Marland, August 2019). A fixed fraction of 6.7% of liquid fuels is assumed to be 'stored' (not oxidised) following table 8 of Marland and Rotty (1984).

For national emissions, apparent gross energy consumption is calculated from production plus imports, less exports, less
supply to international bunkers, and adjusted for changes in stocks. While CDIAC used to use estimates of oxidation rates also for national estimates (Marland and Rotty, 1984), UN data at the national level subsequently improved and non-fuel uses have been removed from national estimates according to reported data since the 2009 edition (pers. comm., Dennis Gilfillan, January 2020), but only for liquid fuels. This means that emissions from, for example, natural gas used as a feedstock in fertiliser production are included, but that those from oxidation of, for example, petroleum coke used as a



chemical reagent may not be. Originally CDIAC converted national energy data in original units for all three fuels directly to emissions using carbon contents from table 13 of Marland and Rotty (1984), but the UN subsequently began to report country-provided energy contents for solid fuels as part of their energy data, and these are now used.

For both global and national emissions estimates, carbon contents from table 13 of Marland and Rotty (1984) are used for gaseous and liquid fuels, while for solid fuels two separate carbon contents are used for hard coal and others (pers. comm.,

Dennis Gilfillan, January 2020).

For cement, production in tonnes from the USGS is multiplied by 0.136 g C/g cement (Boden et al., 1995). A number of authors have raised questions about the accuracy of CDIAC's cement emissions estimates, particularly for China (e.g., Lei, 2012;Ke et al., 2013;Liu et al., 2015). Andrew (2019) discussed the reasons for this method producing inflated emissions estimates, and CDIAC are actively pursuing solutions (pers. comm., Gregg Marland, August 2019).

Other emissions from decomposition of carbonates are not included, but EDGAR data indicate that process emissions in cement production amounted to 78% and 80% of global carbonate emissions in 2014 and 2018, respectively. Put another way, non-cement carbonate emissions make up about 1.3% of global fossil $CO_2$ emissions.

According to Marland and Rotty (1984), emissions from flared natural gas starting in 1971 are calculated from data provided by the US Department of the Interior and Department of Energy, while earlier estimates are taken directly from Rotty

(1974). Rotty (1974) had access to data on flared gas in the USA from 1935, but for other countries used a regression approach based on quantity of oil produced. CDIAC also assume that vented natural gas is oxidised within the same calendar year because of a lack of data separating flared from vented gas (Andres et al., 2012).

Andres et al. (2012) reported uncertainty on global emissions as ±10% at 95%/2sd, while this was subsequently updated by Andres et al. (2014) to ±8.4% at 95%/2sd.

**5.5    The Emissions Database for Global Atmospheric Research (EDGAR)**
The Netherlands Environment Agency (PBL) published the first version of EDGAR in 1995, limited to emissions from aviation, spatially distributed on a 5°x5° grid for the year 1990 (Olivier, 1995). Version 2.0 was published the following year on a 1°x1° grid for 1990, with sectoral, grid and per-country data (Olivier et al., 1996;Olivier et al., 1999).

EDGAR is developed and maintained by the Joint Research Centre of the European Commission, with continued input by

PBL, and is used by the IPCC (e.g., Blanco et al., 2014). The methodology is fully documented by Janssens-Maenhout et al. (2019), describing v4.3.2, presenting emissions for 1970–2012.

The EDGAR $CO_2$ emissions database is released in more than one format, with the fully disaggregated dataset updated less frequently. A 'Fast Track' version is produced every year, partly using extrapolation based on activity data, and is released at a much more aggregated level of detail. The most recently published version at time of writing was v5.0_FT2018, used by

Crippa et al. (2019), for which the publicly available version includes total fossil $CO_2$ (fossil fuels and carbonates) for five sectors, 208 countries plus bunker fuels, for 1970–2018. The more complete v5.0 provides estimates for 35 sectors, 231 countries plus bunker fuels, for 1970–2018; data are not provided by fuel type.



Emissions from fossil fuels are derived using the IPCC Tier 1 approach according to the 2006 IPCC Guidelines from the IEA's energy data (v4.3.2 used IEA's 2014 edition, while v5.0_FT2018 used the 2017 edition), and for this reason they are
identical to those in IEA's emissions database in most years, noting that – for reasons of timing and consistency – EDGAR's release relies on previous years' energy data releases from IEA, and later years are extrapolated using growth rates from BP data (pers. comm., Monica Crippa, January 2020) . Carbonate emissions are largely based on production data from the USGS in addition to extrapolation using activity data such as crude steel production (Crippa et al., 2019).

EDGAR is one of two global $CO_2$ datasets (the other being CEDS) that include global estimates of emissions from all
carbonate decomposition, not just cement production, and thereby covers all fossil carbon sources.

EDGAR's global emissions estimate is calculated as the sum of all country estimates.

EDGAR estimates uncertainty on $CO_2$ emissions by assigning 2σ values that vary by country/region and time (Table 2b in Janssens-Maenhout et al., 2019), and this is in active development. Global uncertainty on $CO_2$ emissions, assuming all errors are independent, is calculated to be ±9% at 2σ.

**5.6    US Energy Information Administration (EIA)**

The EIA is a federal statistical agency formed in the wake of the energy crises of the 1970s to collect and disseminate energy information (EIA, *no date-b*). As with BP and IEA, its primarily concern is with energy, but it thereby has all the data required to produce estimates of $CO_2$ emissions from energy consumption. EIA collect international energy data directly from a large number of sources (EIA, *no date-a*).

The dataset 'Total Carbon Dioxide Emissions from the Consumption of Energy' is part of their International Energy Statistics (EIA, *no date-c*). The product was in 'Beta' from May 2015 through January 2020, when the beta designation was removed (EIA, 2020), but it is not clear what update frequency is intended. At time of writing the data covered the period 1980–2016 for 228 countries and territories.

In the Beta version, the EIA's emissions estimation methodology was stated to be documented in a 2008 report (EIA, *no*
*date-d*), although this document specifically deals with emissions in the US, and makes no mention of international emissions (EIA, 2008). The documentation of the 2020 version of the international dataset at time of writing is incomplete, but appears also to describe only estimates for the US, and there appear to be some problems with the dataset (see Supplementary Information). EIA are working towards a new process that will include more transparency and closer links with the data used in its International Energy Outlook (pers. comm., Perry Lindstrom, March 2019).

The EIA's international energy dataset does not report non-fuel uses, so the (Beta) methodology for estimating emissions first adjusts for these. Carbon in natural gas used for manufacturing nitrogenous fertiliser is assumed emitted, otherwise all non-fuel use is assumed to be sequestered. Coke use in metallurgy is taken to be combusted, rather than a non-fuel use.

In the Beta release, national emissions included bunker fuels along with flared natural gas and vented $CO_2$, with emissions from flared gas reported separately. In the new version released in 2020, emissions from flared natural gas are no longer
reported separately, and are included in the 'natural gas' emissions category.

The EIA provides no quantitative assessment of uncertainty associated with its emissions dataset.



### 5.7 Global Carbon Project (GCP)

The GCP is an international collaboration whose main purpose is to understand the global carbon cycle. Since 2005 it has released a Global Carbon Budget, later to become an annual publication whose release is usually timed to coincide with the UNFCCC Conference of the Parties, and one component of this publication is a fossil $CO_2$ emissions dataset (Friedlingstein et al., 2019).

GCP's fossil $CO_2$ dataset is based primarily on CDIAC, because of that dataset's wide use in the carbon-cycle community for many years. In addition, GCP prioritises data from the Annex-I countries' inventory reports to the UNFCCC, overwrites cement emissions from Andrew (2019), and uses energy growth rates from BP and the USGS to extend time series where applicable.

Combining inventory data with CDIAC for Annex-I countries (38% of $CO_2$ global emissions in 2017) introduces problems of consistency of system boundaries. That which CDIAC includes as emissions for each fuel category is not the same as that which the IPCC Guidelines indicate for the Energy sector (see Section 5.4). GCP uses official inventory estimates for national total (all sectors) $CO_2$ emissions, and maps these to CDIAC's reported system boundaries to reduce inconsistencies and discontinuities (Friedlingstein et al., 2019). For example, GCP's natural gas estimates include not only combustion of natural gas but also any use of natural gas that under CDIAC's methodology is assumed to be oxidised in the short-term, such as use as a feedstock in fertiliser manufacture. For liquid fuels, this includes not only combustion of petroleum products, but incineration of plastics; similarly for coal. In addition, GCP includes an 'other' category for $CO_2$ emissions in the CRF reports that are outside of CDIAC's system boundary, such as those in quicklime production, urea application, and combustion of peat.

In some cases, the GCP overwrites other CDIAC data, for example in the case of Norway. In 2019, GCP used the previous year's edition of CDIAC for China and Saudi Arabia because of apparent problems in the 2019 edition of CDIAC. Further, in 2019 GCP recalculated global emissions as the sum of countries' emissions, rather than using CDIAC's global estimates (Friedlingstein et al., 2019).

GCP's data period is (in the 2019 edition) from 1750–2018, with a global projection to 2019. The dataset is released annually and available both as an Excel/CSV download and via web-based interface (Global Carbon Project, 2018). Documentation is updated annually through the "living data" process at the journal Earth System Science Data.

The GCP assesses uncertainty on global emissions to be ±10% at the 95%/2sd level, after Andres et al. (2012), with uncertainty for developed countries at ±10% and for developing countries ±20% (Friedlingstein et al., 2019). GCP reports uncertainties at 68%/1sd level (Friedlingstein et al., 2019).

### 5.8 International Energy Agency (IEA)

The IEA, established in 1974, is an intergovernmental organisation whose membership draws from members of the Organisation for Economic Cooperation and Development (OECD) (Scott, 1994). The IEA's data coverage originally extended only to the OECD countries, and early publications were continuations of publications made by the OECD (Scott, 1994), but in 1994 this coverage was expanded (IEA, *no date-b*) and today it collects energy data for about 150 countries



(Coënt, 2017). Data for remaining countries are obtained from the UN Statistics Division (pers. comm., Francesco Mattion, February 2020).

The IEA currently has about 30 staff dedicated to energy statistics, works directly and iteratively with national energy data providers, and partners with regional energy data organisations (pers. comm., Roberta Quadrelli, December 2019).

There are five annual questionnaires that members of the OECD, EU, and UNECE are obliged to return: one each for coal, oil, natural gas, electricity and heat, and renewables. The completed questionnaires are submitted directly to IEA, UN, and (if European) Eurostat. For coal, which includes peat and oil shale/sands, data are submitted in mass terms along with both net and gross calorific values (NCV/GCV; energy contents); oil data are submitted in mass terms along with energy in NCV; natural gas data are submitted in both volume and energy terms in both NCV and GCV (IEA, *no date-a*).

Questionnaire data are supplemented with information directly from national administrations, and, where necessary, industry (Coënt, 2017). For all other countries, the "commodity balances … are based on national energy data of heterogeneous nature, converted and adapted to fit the IEA format and methodology" (IEA, 2018b, p. I.17).

The IEA has reported global $CO_2$ emissions since the 1997 release of its "$CO_2$ Emissions from Fuel Combustion" report (Bamberger, 2004), calculating emissions directly from energy data in terajoules using (since the 2015 edition) IPCC 2006

Tier 1 methods and default $CO_2$ emission factors from Table 2.2 of the 2006 Guidelines, with both bioenergy combustion and non-energy use resulting in zero emissions. IEA energy and emissions datasets exclude flaring and venting (IEA, 2019d, b).

In the 2006 IPCC Guidelines, emissions from "non-energy uses of fossil fuels" were reallocated to IPPU from the Energy sector (Eggleston, 2008, p. 15). However, some of these emissions are simultaneously non-energy (i.e. chemical use of

carbon) and energy use of fossil fuels, for example the use of coke in iron production. Because of this, and to maintain consistency with previous work, the IEA includes those emissions which would be reported under IPPU but which constitute use of energy in its estimates of emissions from fuel combustion. All such emissions occur in the metal production industries.

There are three variants of IEA's emissions database, all released annually:

- *Detailed estimates (IEA, 2019a, b)*: includes emissions from 1960 (for OECD members, and 1971 for non-members), 148 countries/35 regions, 47 products, 41 flows. Paid service.
- *2006 Guidelines (IEA, 2019b)*: includes emissions from 1970 (for OECD members, and 1971 for non-members), 185 countries/regions, 5 products, 13 flows. Paid service.
- *$CO_2$ Highlights (IEA, 2019c)*: includes emissions from 1971, 143 countries/24 regions, 6 flows (total fuel combustion
emissions by the sectoral approach, emissions by coal, oil, gas, and emissions from aviation and marine bunkers). A limited sectoral breakdown is available for the most recent data year. Freely available, usually released in October/November.

The IEA has three main definitions for total emissions (IEA, 2018c) (see Figure 2):

- $CO_2$ Fuel combustion: Both that which would be included in IPCC category 1A (Energy: Fuel combustion), and any fuel
combustion in IPPU. This is the IEA's headline definition.




- $CO_2$ Sectoral Approach: Fuel combustion only in IPCC category 1A (Energy: Fuel combustion). This excludes certain emissions in the metals industry.
- $CO_2$ Reference Approach: Fuel combustion calculated using energy supply data. These therefore include some fugitive emissions (e.g. from refineries), in addition to differing by statistical differences (the difference between supply-side and demand-side data). In general IEA expects their Reference Approach emissions to be an overestimate of fuel combustion emissions (IEA, 2018c).

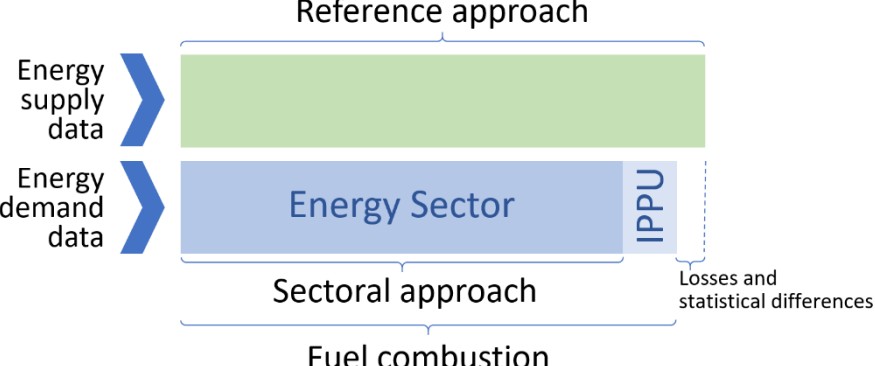

**Figure 2: Schematic explanation of IEA's emissions estimates totals. IEA reports emissions from fuel combustion, using a Reference Approach, based on energy supply-side data (production, trade, stock changes), and a Sectoral Approach, based on energy demand-side data (largely sales). The latter is also subdivided between emissions that fall under the IPCC's Energy Sector and those that fall under the Industrial Processes and other Product Use (IPPU) sector (largely in the metal industry).**

The process of data harmonisation and training actively driven by the IEA means that other organisations that collect directly from national agencies obtain better quality data, benefitting from the IEA's efforts.

IEA's global emissions estimate is calculated as the sum of all country estimates.

The IEA provides no quantitative assessment of uncertainty associated with its emissions dataset.

### 5.9   PRIMAP-hist

The Potsdam Real-time Integrated Model for probabilistic Assessment of emissions Paths (PRIMAP) historical emissions dataset (PRIMAP-hist), led by Potsdam Institute for Climate Impact Research (PIK) is constructed based on a prioritization scheme from other emissions datasets (Gütschow et al., 2016).

For fossil $CO_2$ emissions, data sources include CDIAC, EDGAR, BP, and the CRFs for Annex-I Parties to the UNFCCC. In addition, some emissions for non-Annex I parties were obtained from the UNFCCC's "Detailed data by party" web interface, along with selected biennial update reports. Unique to this dataset, data from non-Annex I parties' Biennial Update Reports are included. Emissions from international marine and aviation bunker fuels are not included.

Version 2.0 was released in January 2019 and includes emissions for 1850–2016, with updated data sources in addition to the use of cement emissions from Andrew (2019). The sector disaggregation is updated to follow the 2006 IPCC Guidelines, and an additional time-series is included that prioritises third-party data over country-reported data (Gütschow et al., 2019a).

Version 2.1 is a minor update, released in September 2019 (Gütschow et al., 2019b).





### 5.10    UNFCCC: CRFs

Since 1996, Annex I parties to the UN Framework Convention on Climate Change (UNFCCC) have been required to submit

emissions inventories for at least the period 1990 to two years before the submission year, following a decision made at the first Conference of the Parties (COP) (UNFCCC, 1995). At the fifth COP in 1999, it was agreed that Annex I parties would report quantitative inventories via a detailed Common Reporting Format (CRF) in Excel format, due 15 April every year (UNFCCC, 2000). In 2019, 45 countries submitted, in addition to the EU's combined submission, since the EU is also a party to the UNFCCC in its own right.

At the fifth COP in 1999 it was also agreed that Annex I Parties' inventories would be reviewed annually by expert review teams, primarily a desk-based review, with in-country reviews every five years (UNFCCC, 2000). Inventories are assessed according to five general criteria: transparency, consistency, comparability, completeness, and accuracy (UNFCCC, 2014). Activity data are compared with data from "relevant external authoritative sources" (p. 112, UNFCCC, 2000), which include International Energy Agency, Food and Agriculture Organization of the United Nations, World Bank, Montreal Protocol,

and the UN Statistical Division (Olsson, 2008).

CRF data files are generated using software developed by the UNFCCC Secretariat following the structure presented by the IPCC Guidelines (see section 4), but are in a relatively poor format for comprehensive analysis, with the 2019 edition's dataset spread over 132,050 spreadsheets in 1,390 separate Excel files (UNFCCC, 2019a). However, some data are available through the UNFCCC's data interface (UNFCCC, *no date*), and Jeffery et al. (2018) have produced a flat-record format

dataset from the Excel files, facilitating more widespread analysis. Revised inventories are typically submitted several times through the year as corrections are made. Revisions from year to year of historical estimates do sometimes result in significant changes (see Supplementary Information).

Energy emissions are estimated using a bottom-up, demand-side, energy consumption approach, called the Sectoral Approach. In addition, a Reference Approach, using coarser, national-level, supply-side data is used to produce a cross-

check. The Reference Approach serves as a quality check using somewhat independent data and a simplified methodology. Parties are required to compare the two approaches in their inventory report (UNFCCC, 2014). In countries that take part in the EU Emissions Trading Scheme (ETS), considerable data is sourced from detailed company-level ETS reporting.

Non-Annex I parties (parties to the UNFCCC that are not listed in Annex I of the Convention treaty text) are requested to submit National Communications (NC) and Biennial Update Reports (BUR), for which the requirements are less stringent

than the CRF. In particular, most of these reports do not include time-series of emissions, but rather a single year.

Each Annex I party estimates uncertainties associated with each emissions estimate, and aggregated uncertainties for totals, following the uncertainty guidelines provided by the IPCC (Frey et al., 2006).

Henceforth in this work I will analyse Annex I emissions inventories and refer to them as 'CRF'.





## 5.11  Other datasets

### 5.11.1  HYDE

The IMAGE 2 'hundred year' (1890-1990) data base of the global environment included historical $CO_2$ emissions for 13 regions (Klein Goldewijk and Battjes, 1995, 1997), but the $CO_2$ data are no longer updated (pers. comm. Kees Klein Goldewijk, February 2019). It calculated emissions for 1860–1949 directly from energy production data (Etemad and Luciani, 1991). HYDE's documentation makes no mention of adjusting for energy trade, which Etemad and Luciani (1991) did not include. For the period 1950–1990 emissions were taken directly from Marland et al. (1994).

### 5.11.2  MATCH

The Modelling and Assessment of Contributions to Climate Change (MATCH) expert group was established by the UNFCCC in 2001 to generate a historical emissions time series in the wake of Brazil's proposal to include historical emissions in negotiations (Höhne et al., 2011;MATCH, *no date*). It was updated by den Elzen et al. (2013) with data from EDGAR, but is no longer updated.

The dataset included emissions from Energy and industry ($CO_2$, $CH_4$, $N_2O$), Agriculture and waste ($CH_4$, $N_2O$), Land use change and forestry ($CO_2$). Emissions were collated from other datasets with the following order of prioritisation:

- UNFCCC submissions (Annex I: 1990–2004, Non-Annex I: 1994 and earlier where available)
- IEA $CO_2$ from fuel combustion 1970–2004 + cement emissions from CDIAC
- US EPA 1990–2005 for $CH_4$ and $N_2O$
- CDIAC 1751–2003
- EDGAR/HYDE 1890–1990, all sectors, 17 regions
- MNP/RIVM IMAGE 2.2: 1970–2100, all gases, all sectors, 17 regions

## 5.12  Summary of selected data sets

Figure 3 provides an overview of some general characteristics of six global emissions datasets. Primary source indicates whether or not the dataset relies on other emissions datasets at all, or whether all emissions are entirely derived from activity data; 'Reports bunkers separately' refers to whether emissions from international bunker fuels are explicitly reported or included in totals; By fuel type indicates whether the dataset publicly reports a breakdown by fuel type (e.g. solid, liquid, gas).



| | Primary source | Uses IPCC emission factors | Includes venting & flaring | Includes cement | Includes other carbonates | Reports bunkers separately | Non-fuel use based on | By fuel type | By sector | Includes official estimates |
|---|---|---|---|---|---|---|---|---|---|---|
| CDIAC | yes | no | yes | yes | no | yes | National data | yes | no | no |
| BP | yes | yes | no | no | no | no | National data | no | no | no |
| IEA | yes | yes | no | no | no | yes | National data | yes | yes | no |
| EDGAR | yes | yes | yes | yes | yes | yes | National data | no | yes | no |
| EIA | yes | no | yes | no | no | no | US data | yes | no | no |
| GCP | partial | no | yes | yes | no | yes | National data | yes | no | yes |
| CEDS | partial | no | yes | yes | yes | yes | National data | only for global | yes | yes |
| PRIMAP-Hist | no | no | yes | yes | yes | yes | National data | no | yes | yes |

**Figure 3: Comparison of some important general characteristics of seven global emissions datasets, with green indicating a characteristic that might be considered a strength.**

## 6    Comparison of emissions datasets

In this section I turn to a quantitative comparison of the major datasets reported above, and exploration of the specific

reasons for those differences. Data used to generate figures 4–19 are available at https://doi.org/10.5281/zenodo.3687042 (Andrew, 2020).

### 6.1    Global Emissions

Total global $CO_2$ emissions across the datasets range from 32.4 Gt to 36.3 Gt in 2014, the most recent year for which all datasets are represented (Figure 4). But these datasets have varying system boundaries beyond the Energy sector, with

inclusion of carbonate emissions varying from none to all, and inclusion of non-energy fossil-fuel emissions varying from some to all. In addition, three datasets include the small amounts of $CO_2$ emitted in the Agriculture sector.



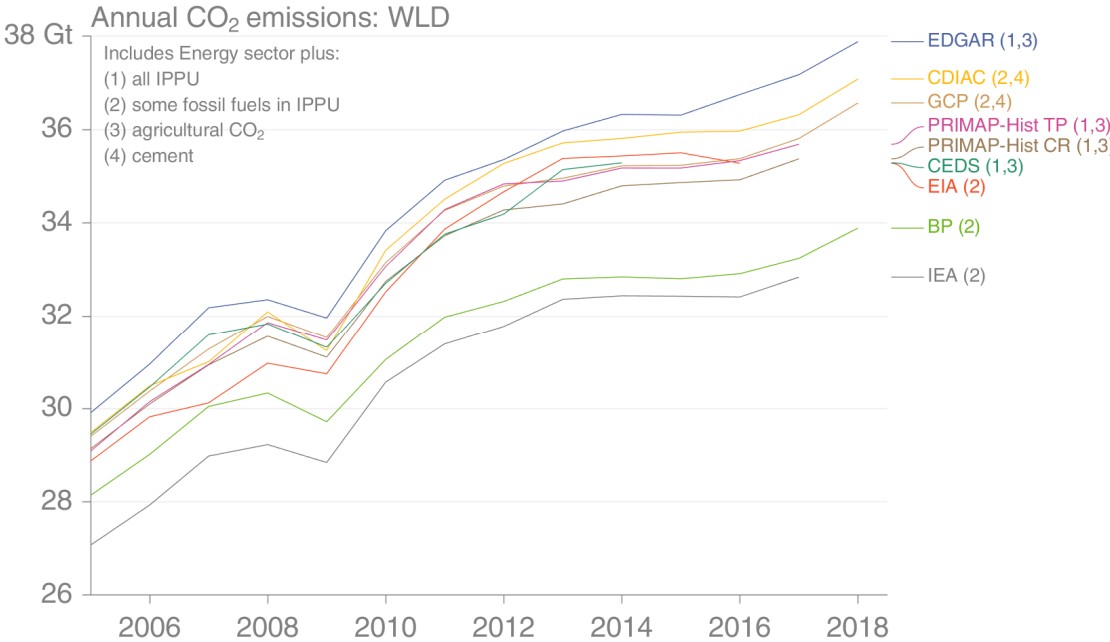

**Figure 4: Annual global CO₂ emissions from various sources: total emissions.**

By selecting subsets of data, it is possible to bring the system boundaries closer together (Figure 5). This reconciliation is
limited by the break-downs available in each dataset: with CDIAC and GCP, cement emissions are removed; with EDGAR,
CEDS and PRIMAP-Hist, IPCC sectors 1, 2B, 2C, 2D, and 2H are selected[1]; while BP, IEA, and EIA do not permit further
adjustment. But with this adjustment to more similar system boundaries, the datasets range in 2014 from 32.4 Gt to 34.2 Gt,
with EIA a clear outlier at 35.4 Gt. Thus, the range of these emissions estimates declines from 3.9 Gt to 3.0 Gt, or to
considerably further to 1.7 Gt if EIA is considered an outlier.

---

[1] Sector 2A includes only emissions from carbonate decomposition, and sectors 2E, 2F, and 2G include only non-CO₂
emissions; 2H was only present in CEDS and PRIMAP-Hist.

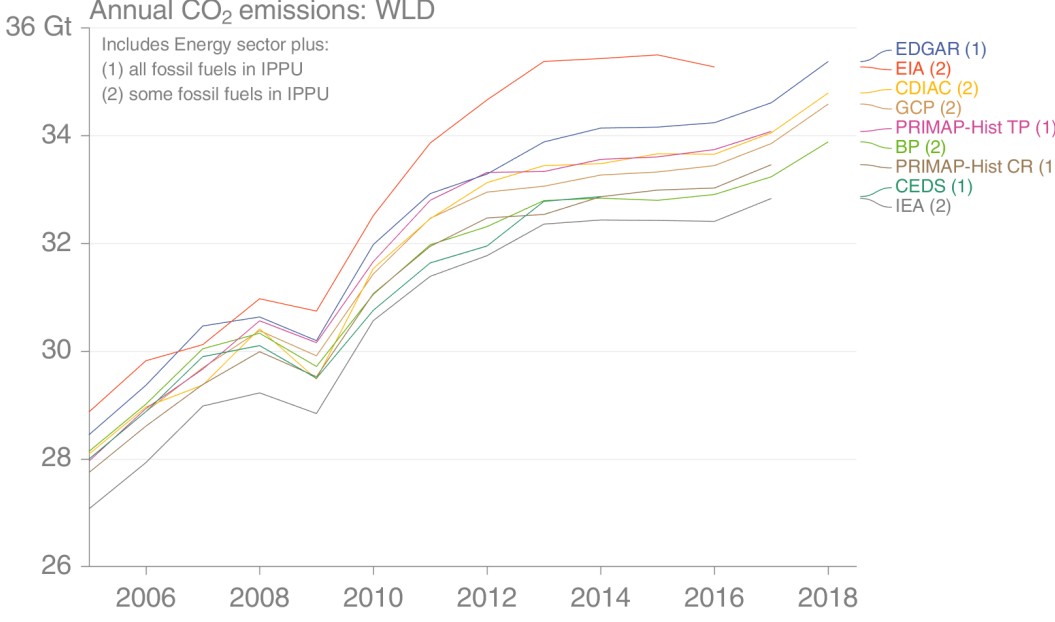

**Figure 5: Annual global CO₂ emissions from various sources using similar system boundaries.**

To explore these differences in detail, Figure 6 compares global emissions in the year 2014 by fuel type across those datasets that provide such a break down, in addition to more aggregated emissions from BP and EDGAR. For BP, the method description allows for emissions from natural gas to be calculated from BP's energy data, but the data for solid and liquid fuels are insufficiently disaggregated to allow replication of BP's emissions calculation method for those fuels. The year 2014 is chosen to maximise the number of datasets that can be compared. Fugitive emissions are those from venting of $CO_2$ and flaring of methane.

For solid fuels, emissions from CDIAC, GCP, and IEA agree to within 0.2 Gt, but EIA's are substantially higher, requiring further investigation (Section 6.1.1). CEDS' emissions from solid fuels are very low as a result of this dataset only including 'Energy' emissions in the breakdown by fuel type: process emissions such as use of coal in the iron and steel industry are included in 'Others'. The sum of solid and liquid fuel emissions in BP is similar to those of CDIAC, GCP, and IEA.

There is a large spread of about 1.1 Gt in emissions from liquid fuels, from 10.9 Gt in IEA to 12.0 Gt in EIA and CDIAC. CDIAC and GCP are expected to have higher emissions in liquid fuels that IEA because they include additional emissions from liquid fuels not used as energy sources, but GCP sources some emissions from Annex I parties' official reports: for Annex I countries, emissions from liquid fuels are about 360 Mt $CO_2$ higher in the CRFs than in IEA, and GCP's reallocation of emissions in Annex 1 countries' IPPU sector to liquid fuels contributes about a further 210 Mt (Figure 8). CDIAC uses energy production data to estimate global emissions, and these are higher than energy consumption data





https://doi.org/10.5194/essd-2020-34


because of statistical differences, particularly in liquid fuels, where the sum of CDIAC's country-level estimates comes to 10.4 Gt $CO_2$ in 2014, much lower than its global estimate of 12.0 Gt.

Emissions from natural gas range from 6.3 Gt in BP to 7.0 Gt in EIA. The somewhat higher emissions in EIA, CDIAC and GCP result from inclusion of eventual emissions from use of fertilisers derived from natural gas, and possibly other feedstock uses of natural gas that result in $CO_2$ emissions. EIA also includes flared (fugitive) natural gas emissions in its natural gas category, contributing about 0.2 Gt.

CDIAC includes only cement emissions in carbonates, but according to Andrew (2019), these are inflated. Since CEDS uses

cement emissions estimates directly from CDIAC, these are also inflated, and its Others category also includes oxidation of all fossil fuels used as feedstocks. EDGAR's v5.0 release introduced an error in cement emissions, unintentionally inflating them by more than 20% for the years 2016–18 (pers. comm., Jos Olivier, December 2019). BP, EIA, and IEA do not include emissions from carbonates.

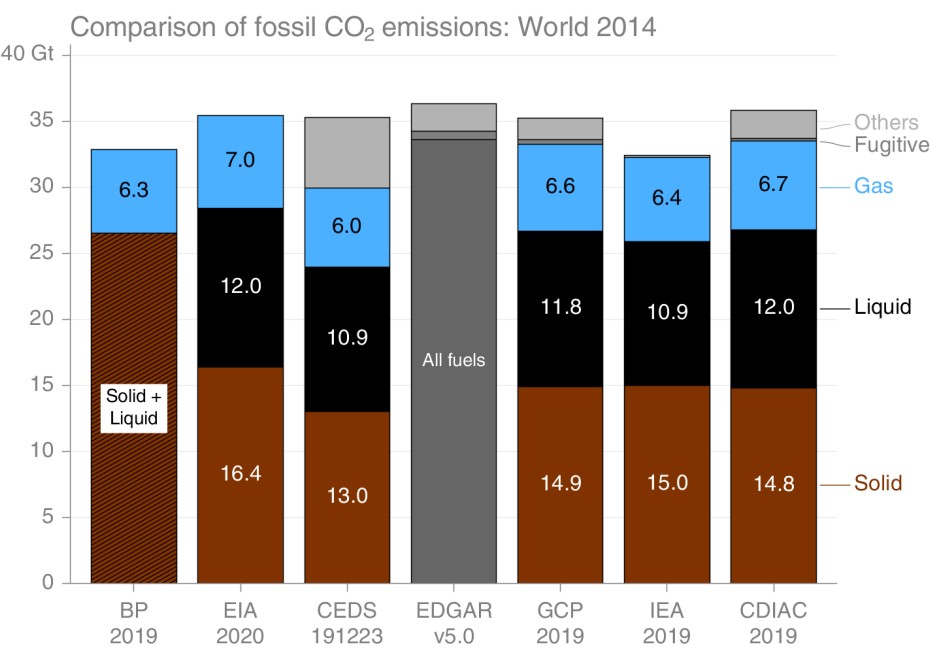

**Figure 6: Comparison of global CO₂ emissions for the year 2014 from seven datasets, broken down by fuel type. BP and EDGAR do not report emissions by fuel type.**

### 6.1.1 Underlying Energy Data

All $CO_2$ emissions estimates are ultimately largely derived from energy data, and this is therefore a key potential source of deviation. It is also a large potential source of uncertainty, relying as it does on national administrations to report correctly.

Figure 7 compares global energy consumption from three energy datasets, demonstrating general coherence but with some deviations. These three datasets are used to estimate emissions by BP, EIA, CEDS, EDGAR, and IEA (see Figure 1). CDIAC and GCP rely on UN energy data, which are likely to be very similar to IEA energy data but are not freely available.



To make the three datasets comparable it was necessary to convert them all to the same units, EJ of net calorific value (NCV). EIA present all their energy data in gross calorific value terms (also known as the higher heating value), and to 390 convert to NCV I use the three simple factors suggested by the IPCC: coal 0.95, oil 0.95, gas 0.90 (Eggleston et al., 2006). The figure shows a significant deviation of EIA's coal energy data after 2010 from the other two sources here, and something of a diverging oil trend between the three after about 2005.

BP's oil consumption numbers lie slightly below those of the IEA and EIA over the entire period.

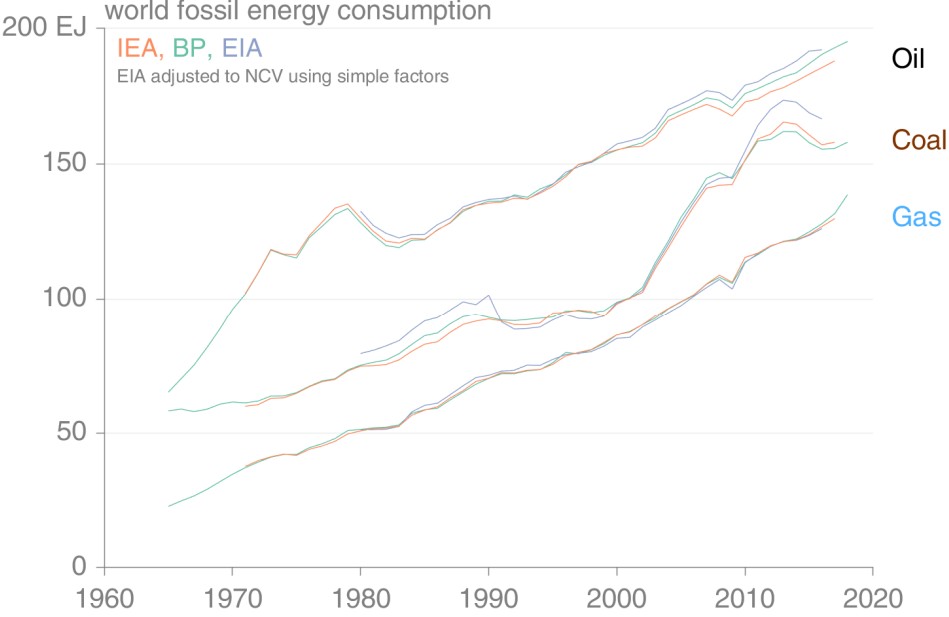

**Figure 7: Global energy consumption estimates from three sources. The EIA reports energy production in Gross Calorific Value (GCV; Higher Heating Value), and these have been converted to Net Calorific Value (NCV) using the standard factors of 0.9 for Natural Gas and 0.95 for Oil and Coal.**

Further analysis shows that the reason for the significant divergence in coal consumption between EIA and the other two sources here is largely in China and appears to result from a difference in treatment of the waste products from washing of 400 coal, which are assumed to have zero energy content by both the IEA and China's National Bureau of Statistics, but not by the EIA (see Supplementary Information for a full analysis). In 2012, EIA's reported coal consumption for China is 9.1 EJ (11%) higher than the IEA's, contributing directly to EIA's estimate of global emissions from solid fuels being 1.5 Gt $CO_2$ higher in that year.

### 6.2 Annex 1 countries

The Annex I parties to the UNFCCC contribute about 40% of global fossil $CO_2$ emissions, and they all submit official estimates of territorial emissions (Section 5.10), allowing comparison of official estimates with third-party estimates. Here I compare the emissions datasets for 38 of these countries, allowing inclusion of eight datasets in the comparison.



The official estimates (CRF) indicate total emissions of 14.0 Gt $CO_2$ in these 38 Annex I countries, excluding emissions from international bunker fuels (Figure 8). The CRF 'Others' category here includes all carbonate emissions as well as non-
energy uses of fossil fuels. GCP's total matches exactly the CRF total, by design. The totals from EDGAR and EIA are also very close, although EIA's estimate excludes emissions from carbonates. EDGAR has almost the same system boundary as the CRFs.

Emissions from solid fuels range between 4.1 Gt in CRF through 4.4 Gt in GCP. GCP reallocates some of CRF's 'Other' emissions back to their original fossil fuels. Both CDIAC and IEA also include some non-energy emissions in this category,
for example from the use of coking coal in the iron and steel industries.

The spread for liquid-fuel emissions is much higher, at 1.1 Gt CO2, with EIA's estimate 24% higher than the IEA's. About 0.5 Gt of this may be that EIA includes bunker fuels in with liquid fuels, and this approximately matches the difference between EIA and GCP. GCP takes CRF estimates and reallocates them, which about 0.2 Gt added to liquid-fuel emissions in this manner. IEA's estimate is over 0.3 Gt lower than the official CRF estimates, more than might be expected, and Section
6.3 demonstrates that this is largely a difference in estimates for the USA.

Emissions from gaseous fuels range from 3.7 Gt $CO_2$ to almost 4.0 Gt. Much of this is a result of the inclusion of emissions from feedstock uses of natural gas, known to be the case in CDIAC, GCP, and EIA. Further, EIA includes fugitive emissions from the flaring of natural gas; the IEA does not include fugitive emissions.

The datasets generally show consistent trends in recent years (Figure SI-30).

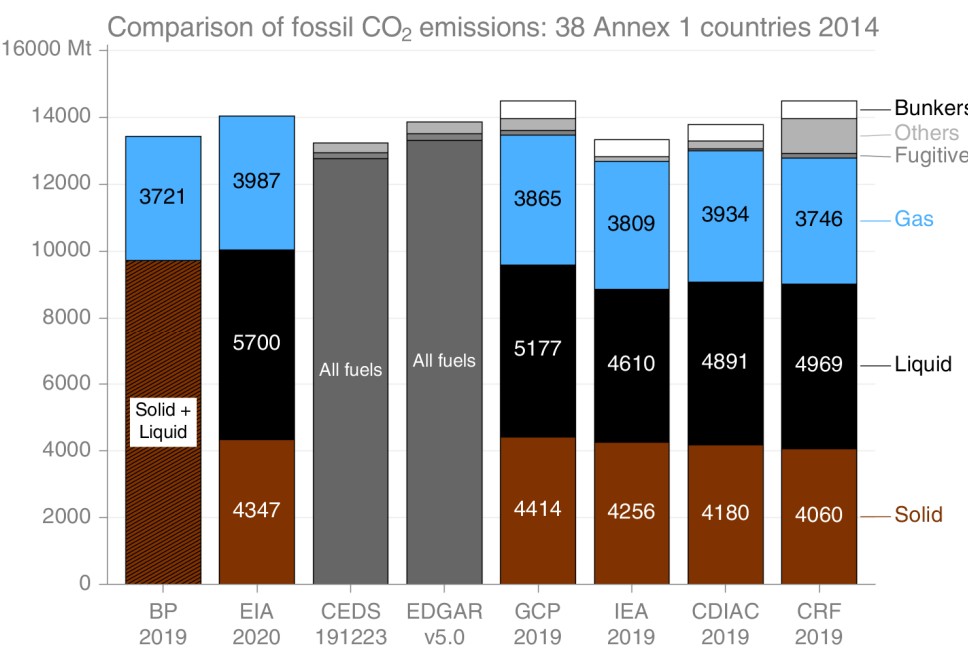


**Figure 8: Comparison of fossil CO₂ emissions in most Annex 1 countries for the year 2014 from eight datasets. Smaller emitters (Cyprus, Israel, Liechtenstein, Malta) are excluded to increase the number of datasets in the comparison.**





### 6.3    USA

Figure 9 compares emissions estimates across eight datasets for the year 2014. Total emissions vary here from 5.0 Gt in

CEDS and IEA to 5.6 Gt in CRF and GCP. It is surprising that CEDS' estimate is so similar to the IEA's, given that CEDS includes all emissions sources while IEA does not, and also given that CEDS uses IEA's energy data as a primary source, but CEDS does not provide a fuel breakdown at country level, hindering further investigation. The CRF, GCP, CEDS, and EDGAR all include all emissions sources, but vary considerably in their total emissions estimates. BP and EIA both exclude emissions from decomposition of carbonates and fugitive sources.

Emissions estimates from solid fuels vary by about 80 Mt, or about 5%, indicating relatively good agreement between the datasets. GCP's estimate is highest, a result of mapping some process emissions in IPPU back to the coal used as a feedstock.

However, emissions estimates from liquid fuels vary by about 350 Mt, or more than 15%. According to the available documentation, EIA's estimate includes bunker fuels, which in the CRF are about 100 Mt. With or without bunker fuels,

there is a large gap between the estimates from EIA, GCP, and CRF and those from CDIAC and, particularly, IEA. Liquid-fuel emissions in GCP are higher than those in the CRF because of reallocation of some IPPU ('Others') emissions to primary fuels, but in the case of the US this makes a difference of only 1%. CDIAC's underlying energy data should be similar to the IEA data, but CDIAC uses different emission factors. The divergence in estimates of emissions from liquid fuels is significant and this warrants further investigation.

Estimates of emissions from gaseous fuels vary by about 50 Mt, or 4%, and part of this is because of the reallocation of the use of natural gas as a feedstock from IPPU back to the gas, known to be the case with both GCP and CDIAC, and also thought to be the case with EIA.

The datasets generally show consistent trends in recent years (Figure SI-31).

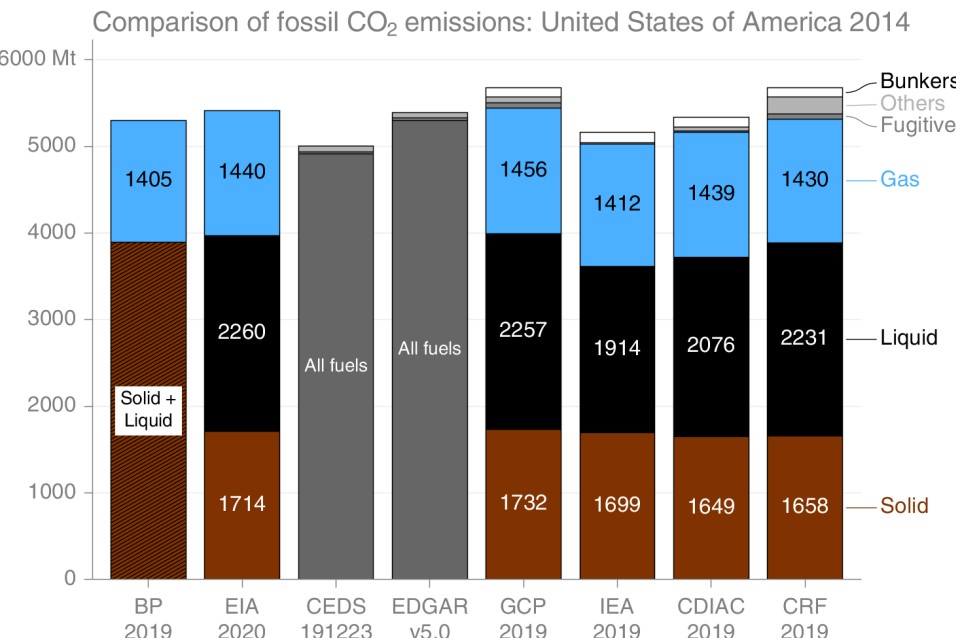

**Figure 9: Comparison of CO₂ emissions estimates in the USA from eight datasets, 2014.**

Figure 10 compares the Sectoral Approach estimates from the CRF and IEA, showing a significant gap between the estimates from liquid fuels. The gap varies over time, and reaches almost 300 Mt, or 15%, in later years.





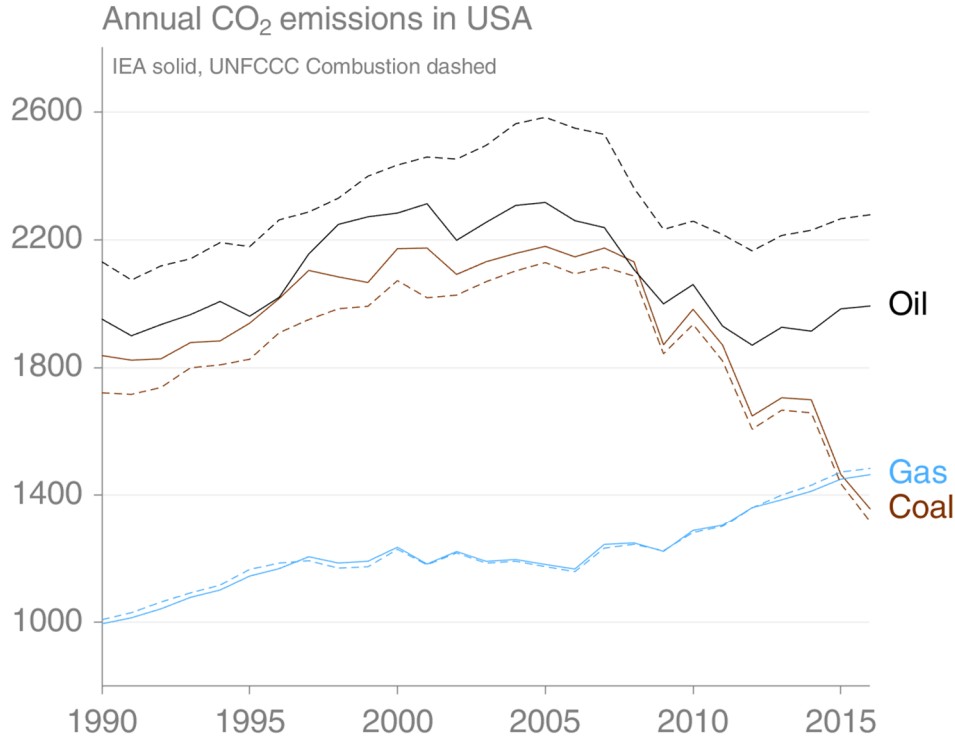

**Figure 10: Comparison of emissions estimates for the USA from IEA and UNFCCC.**

Natural Gas emissions are usually well estimated in all datasets because it usually describes a single energy product, and because energy content exhibits limited regional variation. Oil products, in contrast, are highly variable, with significant variation in energy contents.

Figure 11 focusses on emissions from US consumption of liquid fuels, comparing estimates from the EIA and IEA with those in the CRF (by the US EPA, who submit the CRF). Clearly the approach used (sectoral vs reference) has little to say in

the differences between data sources. The gap between the two 'reference approach' estimates is about 200 Mt CO2 in later years, and between the sectoral approach estimates it is about 300 Mt, as noted above.

Initial analysis (not shown here) indicates several potential reasons for this gap:

- EPA's use of US-specific emissions factors rather than IPCC default factors adds about 20 Mt $CO_2$.
- Inclusion of additional overseas territories in the CRF could add up to 20 Mt $CO_2$, and this is probably largely
limited to liquid fuels.
- EPA's imports of crude oil appear to be 2%–3% higher than those reported by the IEA in energy units. This is probably due to classification methods, such that imports of some commodities are included or excluded in the two datasets as crude oil.
- Use of different energy contents and densities for products: EIA reports most liquid fuel data only in volumetric
units.



- Mapping from EIA's product categories to categories used for emission factors appear to be different. For example, the EIA oil product Motor Gasoline Blending Components is mapped to Other Liquids by EPA but to Motor Gasoline by IEA, and these two products have both different energy contents and different emission factors.

Further resolution of this gap would be possible knowing the conversion factors used by the IEA from EIA's physical units
to energy units, but these were not available.

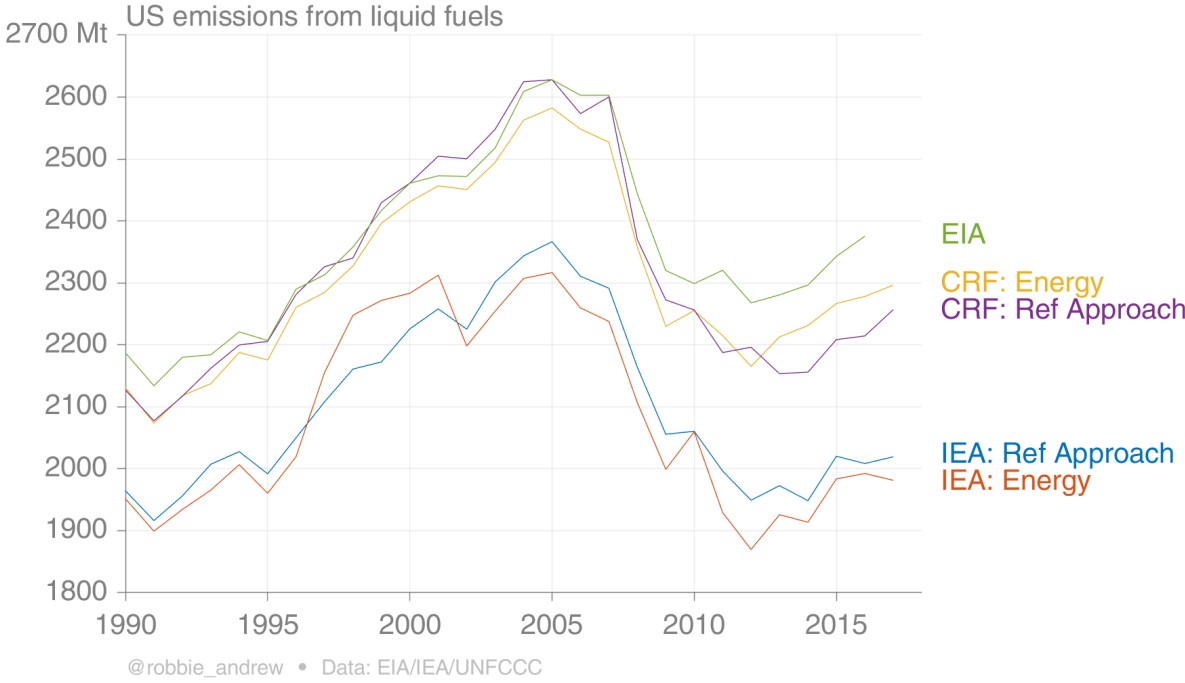

**Figure 11: Comparison of estimates for CO$_2$ emissions from US liquid fuels, from EIA, CRF (EPA), and IEA.**

### 6.4    European Union

For the combined 28 members of the European Union, total CO$_2$ emissions vary across the datasets from 3.2 Gt in IEA to 3.7
Gt in EIA, a range of about 15% (Figure 12). However, EIA is known to include at least some bunker fuels, significantly increasing its estimate of emissions from liquid fuels; BP also includes bunker fuels in its estimates. Looking only at the datasets that include all sources of CO$_2$ emissions, but exclude bunker fuels (i.e. CEDS, EDGAR, GCP, CRF), these exhibit a much smaller range from 3.4 Gt to 3.5 Gt. While the total of GCP is by design equal to that of CRF, and therefore does not provide additional information, the estimates by CEDS and EDGAR are independent and lie very close to the CRF total.

When looking at total emissions from fossil-fuel combustion, those in IEA, CDIAC, and CRF are very close, varying by only 41 Mt out of about 3.1 Gt. As noted previously, GCP's estimates by fossil fuels are larger because they reallocate some IPPU emissions sources back to fuel categories. The 'all fuels' totals for CEDS and EDGAR are all emissions deriving from fossil fuels, both in Energy and IPPU, and these totals are very similar to those in GCP.





Emissions from both solid and gaseous fuels are very similar across datasets. Solid fuel combustion emissions estimates
(excluding GCP) vary by only 26 Mt, about 2.5%. Gas fuel combustion estimates vary by 40 Mt, about 5%, with CDIAC,
GCP and EIA all known to include on-farm emissions from fertilisers made from natural gas, excluded by IEA and CRF
from fuel emissions (IEA excludes these entirely, while CRF reports these in the Agriculture sector, in the figure included
with 'Others').

The variation in emissions from liquid fuels is much larger. However, in the case of GCP this is a result of the reallocation of
IPPU emissions to liquid fuels, and in the case of EIA there are bunker fuels included, increasing both estimates. Where
these factors do not occur, in IEA, CDIAC, and CRF, the range of emissions from liquid fuels is only 53 Mt, about 4%.

There is substantially higher agreement across datasets for the EU than for the USA. One possible explanation for this is that
the data methods used by EU members states lie very close to those used by the IEA, a result of inter-country collaboration
leading to learning and standardisation of methods, coordinated via regular reporting to the European Environment Agency
(EEA). This includes the use of common units of measurement, such that no third party need make assumptions about
energy contents of different fuel types. As a result, the underlying energy data used by each emissions dataset are much more
uniform than those used in generating US emissions estimates.

The differences between datasets in 2014 are consistent across time, apart from EIA, which has diverged since 2006, as
shown in Figure SI-32.

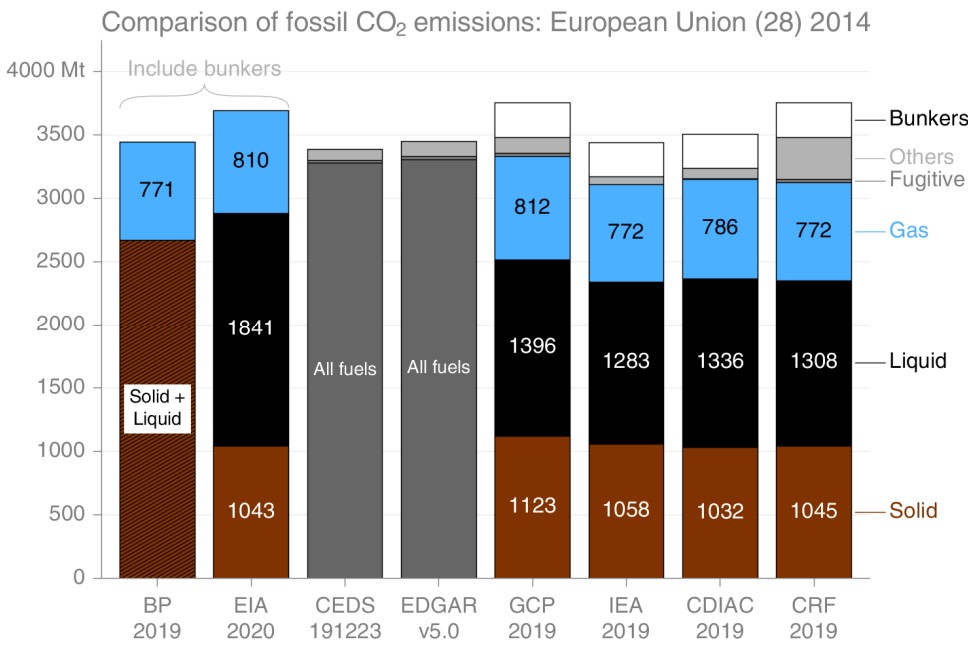


**Figure 12: Comparison of fossil CO₂ emissions from eight sources for the European Union (28 members).**



### 6.5  China

China is not an Annex I party to the UNFCCC, and therefore has different reporting lines and requirements, and furthermore
as an emerging economy is expected to have less institutional experience and capacity in national statistics. It also has a very
large population and a rapidly developing and changing economic structure, making the collection of accurate statistics
across millions of enterprises a significant challenge (Korsbakken et al., 2016). China does not report annual emissions, but
has commenced biennial reporting to the UNFCCC, each report for a single year. In lieu of a CRF, here I compare the global
datasets with the official Second Biennial Update Report (BUR) from China, which presents estimates for the year 2014
(Anon., 2018).

At first glance there is substantial variation in estimates from different datasets, with total $CO_2$ emissions ranging from 9.2
Gt to 10.8 Gt, a variation of about 15%. There are two main reasons for this large variation. The first is that the EIA's
emissions from solid fuels are significantly larger than other datasets, as was discussed in Section 6.1.1, with further details
given in the Supplementary Information. The second is that China has very large emissions from sources other than
combustion of fossil fuels, and system boundary differences therefore explain most of the difference between the lower
estimates (BP and IEA) and the higher estimates.

When looking at total emissions from fossil fuels, and excluding EIA, the estimates vary much less. The Total emissions
from CEDS and CDIAC are very similar to those in the BUR, with EDGAR somewhat higher and GCP a little lower. GCP's
fossil-fuel emissions here are by definition identical to CDIAC, so their similarity does not tell us anything about
independent consistency across datasets, but it uses an independent, lower estimate of emissions from cement production.
While CDIAC's fossil-fuel emissions are very similar to the fuel emissions reported in the BUR, the BUR's 'other'
emissions here include about 410 Mt from non-energy use of fossil fuels. Moreover, CDIAC excludes emissions from
decomposition of carbonates outside of cement production, which have been estimated to be about 450 Mt in 2014 (Cui et
al., 2019). Both CDIAC and CEDS use what are thought to be inflated estimates of China's cement emissions (see section
5.4), but their emissions sources other than fossil fuels are perhaps coincidentally very similar to those in the BUR.

There are only two estimates of emissions from coal, again excluding the EIA: 7425 Mt from CDIAC and 7591 Mt from
IEA, a difference of 166 Mt, or about 2%. For liquid fuels, the estimates range from 1173 Mt in IEwwA to 1500 Mt in EIA,
and it is unclear why the EIA estimate is so high. Reported bunker fuels are surprisingly small in China, given the size of its
international trade. The BUR reports emissions of only 51 Mt $CO_2$ (Anon., 2018), insufficient to explain EIA's high liquid-
fuel emissions. For natural gas, estimates range from 291 Mt in BP to 355 Mt in EIA.

The absolute range of estimates has grown as emissions have grown, and there are some considerably differences in trend in
some recent years (Figure SI-33).

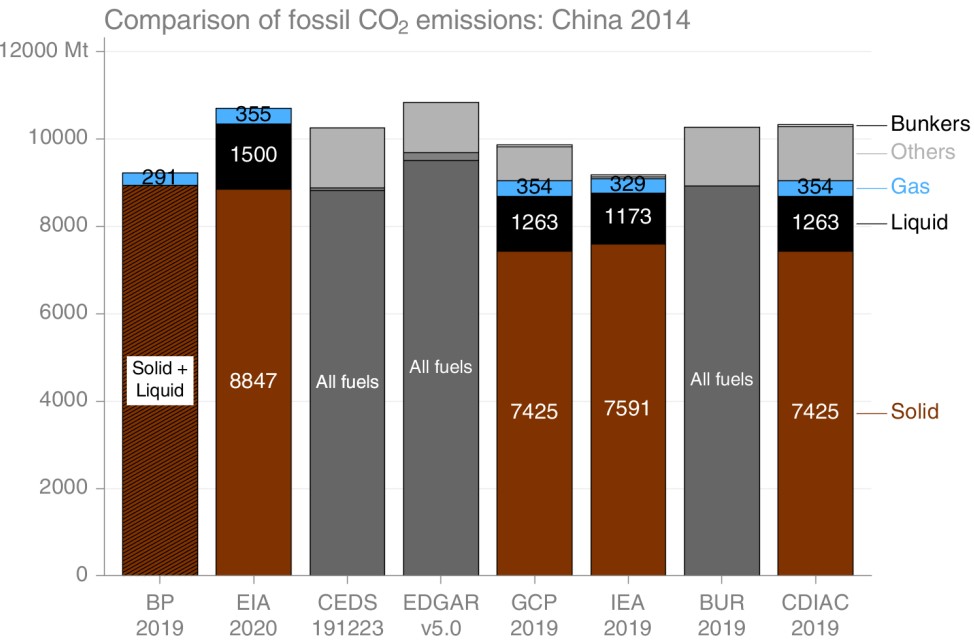

**Figure 13: Comparison of fossil CO₂ emissions from seven sources for China.**

### 6.6 Other illustrative cases

Here I present several cases to compare datasets and explore the specific reasons for differences between estimates. For this I use estimates broken down by major fuel category. As before, the year 2014 is chosen to maximise the number of datasets in the comparison.

#### 6.6.1 BENELUX

The BENELUX group of countries consists of Belgium, Netherlands, Luxembourg (Figure 14). For emissions from solid fuels, the datasets agree well, with a maximum variation of 8% around the median. The outlier is GCP, which re-assigns some of the 'Other' emissions from the CRFs to solid fuels. For liquid fuels there are much larger variations, notably the very high estimate presented by BP, a direct result of this dataset's inclusion of bunker fuels (about 77 Mt) in liquid fuels. The EIA's new (2020) dataset does not appear to include bunker fuels for the Netherlands in 2014, but does for other years (see Figure SI-26). Even so, EIA's estimate for liquid fuels is very high compared to liquid fuels in the other datasets. Among the other datasets, GCP is about 9% higher than the CRFs, again because of re-assignment of some 'Others' to the source fossil fuel. For emissions from gaseous fuels, one can see the more of the consequences of these different mappings, but in addition the mapping of emissions from natural gas used in production of artificial fertilisers mapped to gas emissions for EIA, CDIAC, and GCP, but to Others in the CRFs, and absent in both BP and IEA. The IEA has low 'other' emissions because these include only emissions from the use of fossil fuels as reagents in the iron and steel industry, while CDIAC's low 'other' emissions include only process emissions from cement production. EDGAR and the CRFs include all non-energy emissions in 'other', while GCP includes only emissions from carbonates (including cement production) and maps other

process emissions back to the fossil fuel sources. GCP's total emissions are by design equal to those in the CRF, while EDGAR's use of bottom-up data gives a very close match to the CRF total. CDIAC's total is also very close in this example

to the CRF, and the IEA is slightly lower because of its slightly smaller system boundary, particularly its exclusion of emissions from carbonates. EDGAR does not report country-level emissions from bunker fuels.

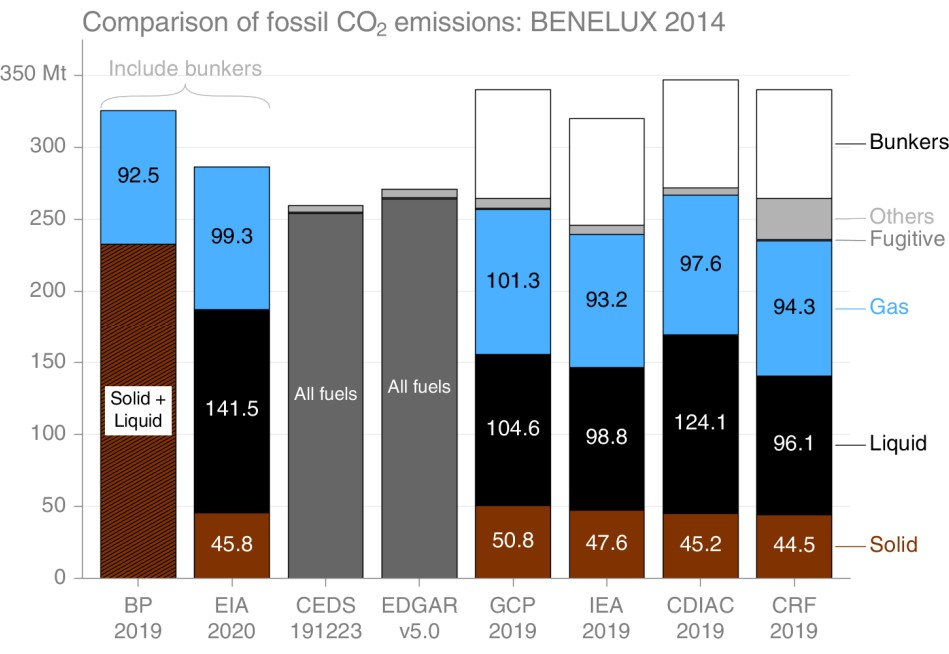

**Figure 14: Comparison of fossil CO₂ emissions from seven sources for the Netherlands, Belgium, and Luxembourg.**

### 6.6.2   Estonia

The next example is for Estonia (Figure 15). For solid fuels there are significant differences, with both EDGAR and EIA having very low estimates. Estonia uses oil shale, an energy-rich sedimentary rock, for about 70% of its primary energy supply, and according to the CRF its combustion contributed about 90% of energy emissions in 2017. IEA energy data indicate that about two-thirds of solid oil shale is used directly to produce electricity, with the remainder converted to liquid and gaseous fuels (IEA, 2019e). The EIA does not include oil shale in its energy or emissions data for Estonia, resulting in a

very low estimate of total emissions.

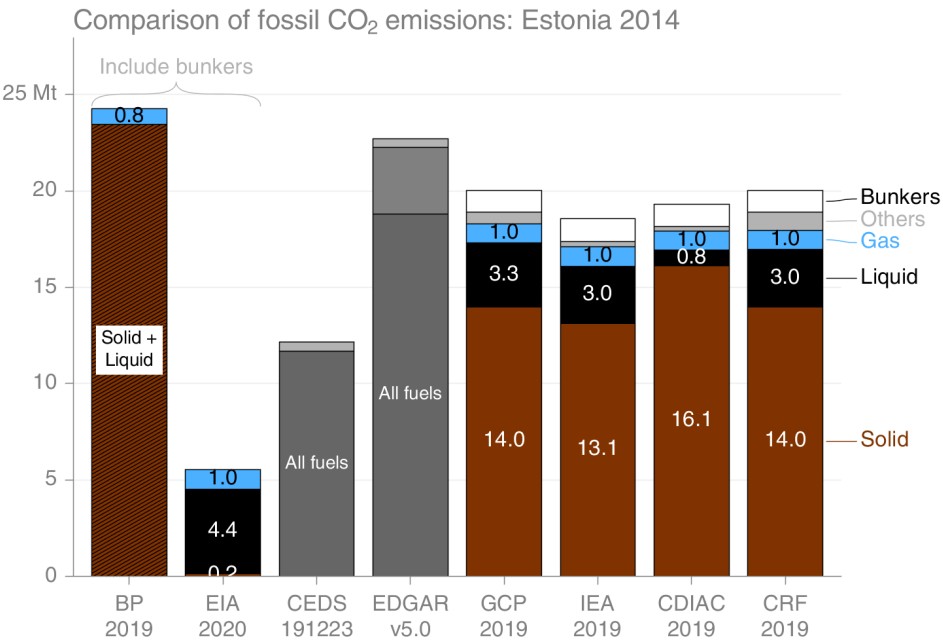

**Figure 15: Comparison of fossil CO₂ emissions from seven sources for Estonia.**

### 6.6.3    Iceland

Iceland has a large metal production industry, resulting in large industrial process emissions from the decomposition of
carbon anodes in aluminium production as well as emissions in iron and steel production, and the CRF indicates that these
make up almost half of Iceland's total CO₂ emissions in 2016 (Figure 16). Non-energy emissions from aluminium
production are excluded from the system boundaries of IEA, CDIAC, BP, and EIA, resulting in substantially lower 'total'
emissions from these datasets (BP does not present data for Iceland, so is not shown on the chart). GCP reassigns these
emissions in the aluminium industry to liquid and solid on the assumption that the carbon used to make the anodes is sourced
20% from coal and 80% from petroleum coke. The EIA, again, includes bunker fuels in their estimate of emissions from
liquid fuels, in the case of Iceland having a substantial effect on total emissions. The fugitive emissions indicated in the CRF
are replicated in GCP, but not present in other datasets, and this is because these emissions are from geothermal energy
generation, which are not included in the other datasets. For example, EDGAR's estimates of fugitive emissions appear to be
based only on fossil fuel production (Janssens-Maenhout et al., 2019).

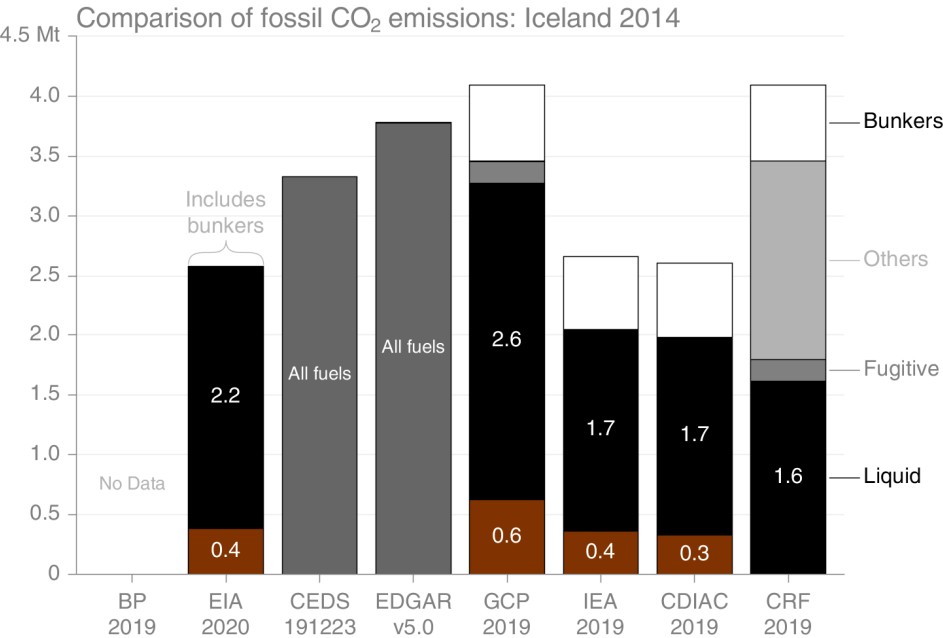

**Figure 16: Comparison of fossil CO₂ emissions from six sources for Iceland.**

### 6.6.4    Norway

Norway produces both oil and natural gas – about 90% of which are exported – and also has a significant metal industry. CDIAC's estimates for emissions from liquid fuels are particularly high, a result of its method being based on the reference approach, calculating apparent consumption from production, net imports, and stock changes, rather than using reported energy consumption. Norway's official estimates using the reference and sectoral approach differ by as much as 45% (Miljødirektoratet, 2019). Norway has repeatedly been asked by the expert review team (ERT) to explain this difference (UNFCCC, 2019c) and has worked since 2011 to reduce the inconsistency. As stated by the IPCC in its guidelines, "for countries that produce and export large amounts of fuel, the uncertainty on the residual supply may be significant and could affect the Reference Approach" (p. 6.13, Treanton et al., 2006a). That the EIA's estimate of liquid-fuel emissions is so close to those from CDIAC strongly suggests that the EIA has also based their estimate on apparent consumption, something that is not documented. The IEA's reported emissions from solid fuels include use of coal as a feedstock in the iron and steel industry, which the CRF reports under IPPU (here 'others'). BP's lower emissions estimate for natural gas results directly from a lower estimate of natural gas consumption, the reason for which is unclear.

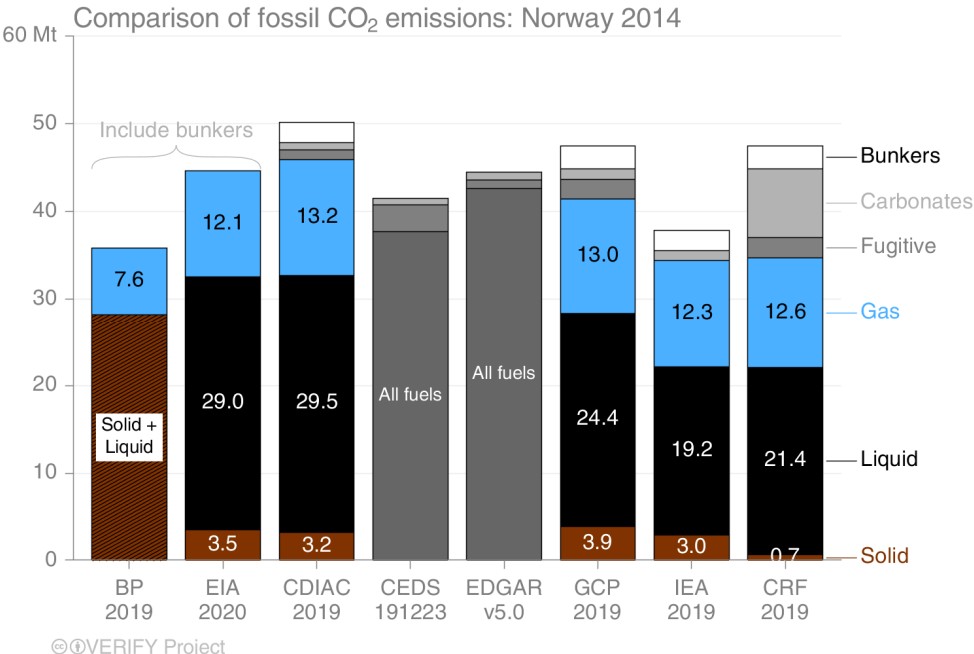

**Figure 17: Comparison of fossil CO₂ emissions from eight sources for Norway.**

### 6.6.5 Nigeria

For Nigeria, there are substantial differences between datasets. Both CEDS and EDGAR have much lower emissions from fuel combustion than the other datasets, and this is because they have used an earlier energy data source, as shown by the addition of the 2017 edition of IEA's emissions dataset for this comparison. Liquid fuel consumption data have since been revised significantly upward.

EIA's emissions from natural gas are much higher than in other datasets due to the inclusion of flaring in natural gas, which for Nigeria makes a substantial difference, and EIA's natural gas emissions here are very similar to the total of gas and fugitive emissions in CDIAC.

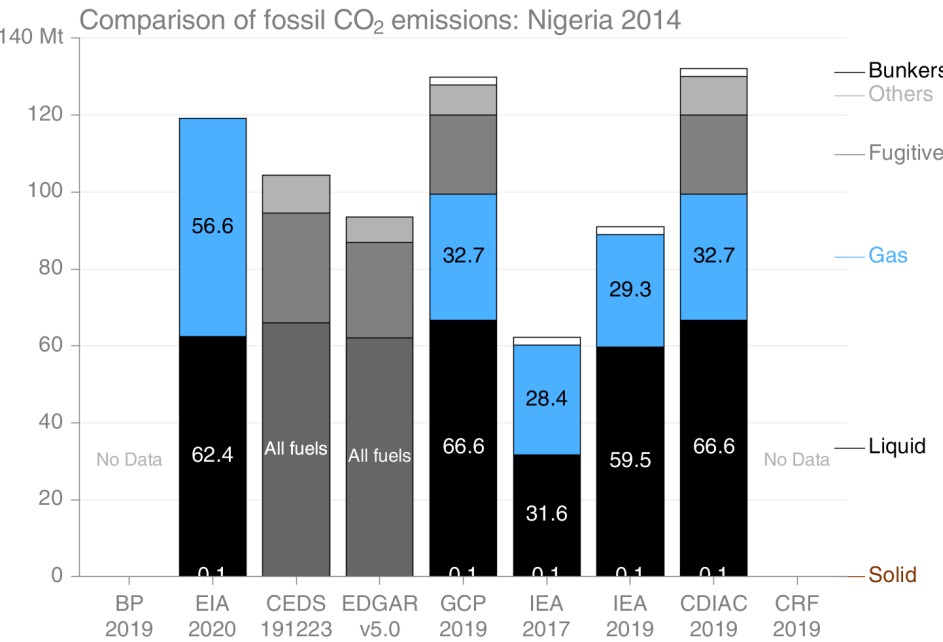

**Figure 18: Comparison of fossil CO₂ emissions from seven sources for Nigeria.**

## 7    Cumulative global emissions

It has been shown that global temperature rise can be expected to be approximately linearly related to cumulative global $CO_2$ emissions through to over 6000 Gt $CO_2$, while cumulative emissions since 1850 are in the order of 2400 Gt $CO_2$, including emissions from land-use change (Friedlingstein et al., 2019;Matthews et al., 2009;Allen et al., 2009;IPCC, 2014). This has been used to determine so-called 'carbon budgets', i.e. the total cumulative $CO_2$ emissions corresponding to specific temperature targets, such as 2 °C (Rogelj et al., 2016). For this it is clearly important to have reliable estimates of cumulative global $CO_2$ emissions.

Figure 19 shows the cumulative difference between global fossil carbon emissions for PRIMAP, CEDS and GCP compared with CDIAC; these are the four emissions datasets that have long time series. I compare with CDIAC because that is the longest standing global emissions dataset, and is used by the IPCC to report cumulative emissions (Stocker et al., 2013), without suggesting that it is more accurate than the other datasets here. The differences between GCP and CDIAC are relatively small, never more than 10 Gt, and result partly from GCP's calculation of global emissions as the sum of countries' emissions while CDIAC uses an independent, global production method. CEDS deviates much more, reaching a peak of about 40 Gt cumulative difference in recent years. Partly this is explained by higher estimates for emissions from bunker fuels, but also emissions in Europe are higher in CEDS 1960–1990, for reasons that are unclear. PRIMAP, whose emissions prior to about 1970 are reported to be based largely on CDIAC, nevertheless diverge strongly from CDIAC's cumulative emissions, reaching a peak of almost 75 Gt cumulative difference. This amounts to about 5.1% more $CO_2$ over this period.


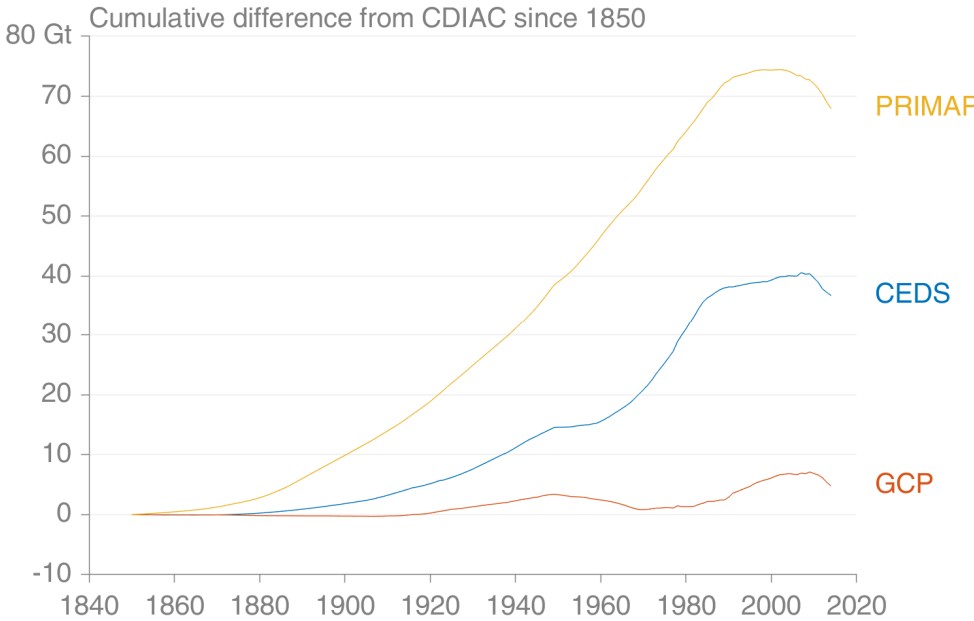

**Figure 19: Difference of cumulative global CO₂ emissions from various datasets from those of CDIAC, starting in 1850.**

## 8    Discussion

In this article I have discussed the historical development of estimating global $CO_2$ emissions and summarised the methods used by the most important current datasets. Given the complexity of the methodologies and source datasets involved, with subtle but important differences, it is unsurprising that documentation in some cases is difficult to follow. In addition, some methods have changed since documentation was published, including that of CDIAC. There is certainly room for improvement in describing the methods and data sources used for construction of emissions estimates.

All emissions datasets build from data on energy and other activities, such as cement production, but it is emissions from fossil fuels and therefore energy data that is most important. There are several major energy datasets with global coverage: IEA, EIA, BP, and UN. While some differences in emissions datasets are due to the use of different energy datasets, different processing methods can result in different emissions estimates even when emissions datasets rely on the same underlying energy data.

Fundamentally, energy datasets are not independent, with all relying on the same underlying energy data in physical units reported by national agencies, with few exceptions. However, differences in interpreting these data, converting to the energy units required for estimating $CO_2$ emissions, emissions factors used, and of course errors, all result in different emissions estimates.

An additional core reason for differing emissions estimates is simply the coverage of the dataset, whether geographic coverage or type of emitting activity. Some datasets do not include emissions from the decomposition of fossil carbonates, such as in the production of cement, while others include international bunker fuels in national estimates. The inclusion of



emissions from non-energy uses of fossil fuels also varies across datasets. For non-fuel emissions, few datasets include estimates of all emissions from decomposition of fossil carbonates.

Close comparison of emissions datasets in this article revealed that estimates of global emissions vary considerably less when they are adjusted to match a common system boundary: which emissions sources are included. Excluding the apparently overestimated estimates from the EIA, and making best efforts to harmonise system boundaries, global emissions

in 2014 varied by 1.7 Gt $CO_2$ (about 5%), and in that comparison residual system boundary differences remained. The simple range of total emissions across datasets is therefore a poor measure of uncertainty. Many of the differences between datasets can be explained from the methods used, the approaches used to allocate to categories, and known system boundaries, and these can make substantial differences between datasets for specific countries, as was shown in the cases of the BENELUX countries, Iceland, Estonia, Norway, and Nigeria.

That said, some discrepancies were found, such as EIA's apparent overestimate of China's consumption of coal energy, and the sizeable differences between estimates of US emissions could not be fully explained. Emissions and energy datasets are complex, and errors are bound to occur, requiring careful checks and comparison with other datasets.

Of the three large emitters investigated here (USA, EU, China), estimates of the European Union's emissions varied the least, after known differences in coverage were accounted for. It is expected that this results from the considerable effort put

into energy and emissions statistics in the EU, combined with close collaboration both between countries and with the EEA and IEA. The EU's Emissions Trading System also acts as a valuable data source.

In contrast, the large discrepancies in estimates of emissions from liquid fuels in the USA warrant further investigation beyond what was possible in this article. The reporting of energy in physical units and use of gross calorific values rather than net calorific values, among other things, hamper a quantitative comparison between datasets.

For China there was relatively good agreement between datasets for emissions from fossil fuels, apart from the EIA, mentioned earlier. While there is good agreement, revisions have previously led to substantial changes in all datasets. This problem isn't limited to China, though, as the example of Nigeria showed.

Little effort has been put into creating independent estimates of emissions prior to 1950, with only two examples known, and no previous comparison of these two found in the literature. Two examples of potentially overestimated emissions in the

early 20[th] century were highlighted. Given the global carbon imbalance in the middle of the 20[th] century, further investigation of emissions estimates in this period is warranted.

Cumulative emissions are important for climate modelling and comparisons with remaining carbon budgets, which tie emissions to temperature targets. Brief analysis here of datasets with long time-series shows considerable divergence, at up to 5% cumulative emissions over 1850–2014. Further analysis of the causes of these long-term divergences is required given

the importance of cumulative emissions estimates.

Much of the difference between different estimates of $CO_2$ emissions results from differences in system boundaries. Given this, and an observed lack of awareness of this fact in the carbon community, it behoves data providers to find a way to be



more explicit about what is and what is not included in datasets. Simply saying 'energy-related emissions', for example, without clarifying what is excluded is likely to leave many readers uninformed.

Reconciling estimates of emissions from different sources is an important goal, but these estimates are much less independent than many believe because the energy data that underlie all estimates originally come from single sources in each country. While it is appropriate to place a certain amount of faith in these agencies, errors and omissions do happen, and independent methods such as atmospheric inversions do have a place in increasing our confidence in the level of global emissions.

The process undergone in this article is in essence one of verification. While 'true' emissions cannot be known, by comparing different datasets methodically, differences that result from system boundaries and allocation approaches can be highlighted and set aside to enable identification of true differences, and potential errors. This must be an important way forward in improving global datasets of $CO_2$ emissions.

## 9    Acknowledgements

The work presented here was funded by the EU H2020 project VERIFY, grant agreement No 776810. I would like to express my gratitude to Jan Ivar Korsbakken (CICERO) for his assistance in investigating the differences between IEA, NBS, and EIA data on Chinese coal; to Roberta Quadrelli, Francesco Mattion and Julia Guyon (IEA) for their assistance in understanding IEA data; to Dennis Gilfillan and Gregg Marland (Appalachian State University) for their assistance in understanding the methodology now used to produce CDIAC data; to Monica Crippa and Efisio Solazzo (JRC) for their

assistance with understanding EDGAR methods; to Steve Smith (PNNL-JGCRI) for assistance in understanding the CEDS database; to Johannes Gütschow (PIK) for assistance in understanding the PRIMAP-Hist dataset; to the VERIFY WP5 team for useful discussions; and to Glen Peters (CICERO) for numerous discussions over the years on the complex details of emissions data. The IEA World Energy Balances, World Energy Statistics, and Detailed $CO_2$ Emissions were used with the kind permission of the IEA, http://www.iea.org/statistics, all rights reserved.

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
