# Peer review of "A comparison of estimates of global carbon dioxide emissions from fossil carbon sources"

_Earth System Science Data, 2020_

## Referee Comment (RC1) · Anonymous Referee #1 · 31 Mar 2020

**A comparison of estimates of global carbon dioxide emissions from fossil carbon sources**

Comparing global $CO_2$ emissions is no small undertaking, but immensely important in the modern era as space-based measurements need ground verification. Robbie Andrew does a thorough job in stating the problem, describing the different boundary conditions of each dataset, and doing a more intensive comparison of selected countries or groups of countries. This is similar to the efforts of Andres et al. 2012 and Macknick 2011, but there is a need for a newer update. Most of my commentary is for copy edits and some clarification of sources and language/terminology, bit I recommend this for publication with minor revisions.

Line by line comments

Page 1, line 24-25: Is there a source for this somewhere?

Line 26: "realm of scientific enquiry"

Page 2, line 7-8: A simple listing of these core reasons would be helpful to the reader

Line 9-10: What exactly is the total budget we are drawing near? More detail would be helpful here, especially in connecting with section 7

Line 11-13: Does this really helps reduce uncertainty, or rather identifies potential sources of uncertainty?

Line 19-22: a list of acronyms used would be helpful for the reader

Page 3, Line 3-7: Regular updates to procedures like this (Comparisons of $FFCO_2$ from different sources) are important, but maybe a connection to the IPCC's 1.5 C report as to why this is especially relevant to address.

Page 4, Line 28-30: What were Keeling's global and cumulative estimates, and how do they compare to the previous estimates? Don't know if that is available, but would be interesting to add.

Page 5, line 13: "estimates had be based"

Page 5, line 29: (REF)?

Page 7 line 19-20: Is there a reason for the separation? Or just a discipline versus generic issue?

Page 14 line 96: Reference needed for USGS minerals yearbook.

Page 15 line 128-129: The estimates of uncertainty needs to be on a consistent format (See also lines 152 and 203)

Page 24, line 363: "in liquid fuels than"

Page 28, line 442-444: Any inclination as to why there is this difference between the other datasets and IEA/CDIAC? Treatment of nonenergy uses maybe? Carbon content of fuel types?

Page 33, line 32: IEA, not IEwwA

Page 34 line 33: "one can see more"

Page 40 line 637: Would a recommendation to data producers be to make their methodology and data sources more transparent to allow for better understanding?

Page 41 line 656: What would a better measure of uncertainty be? This is a good measure of variance in system boundaries and how they relate to outputs though.

Page 42 line 690-693: What about space-based measurements?

---

## Referee Comment (RC2) · Anonymous Referee #2 · 16 Apr 2020

Review ESSD-2020-34

Substantial timely effort to resolve "range = uncertainty" "tangle". Large challenging effort for which time and expertise required remain hidden or at least understated.

Overall, a remarkable effort of immense value. Everyone involved in emissions and budget calculations, as well as larger numbers of folks engaged in numerous (often divergent) political approaches and strategies around lowering those emissions, should read at least the cautionary Discussion (Section 8). Authors and advocates of various emission products, even if their particular product does not appear here, need to learn from this work. If pressed, I would recommend Figure 3 and Figure 5, but specific countries and specific analysts will need to explore carefully through all details of Section 6. I strongly recommend publication in ESSD!

Author several times notes the absence of a definitive reference emission data product. Add a statement, at least once, that extends that statement to the inevitable conclusion that no data product discussed here can claim fully-global all-sector coverage despite their titles. (Perhaps GCP comes closest?) Author several times notes benefit of well-staffed data compiling organizations (e.g. IEA) and of positive outcomes enhanced by training and consultation (e.g. in EU). Bring these positives forward into a recommendation, perhaps near the end of Section 8? Recommendations as they stand (mostly sotto voce) seem to imply more work to tease out differences. Call out at least a few good examples that might serve to minimize errors and differences? Set yourself or colleagues up for a good EU proposal to accomplish some of the needed rectification?

This reviewer remains a bit confused about which data sets the author evaluates and why. Early in the manuscript, the author disclaims attention to gridded emissions products. For good reason - this manuscript focuses on source ("system boundary") uncertainties rather than disaggregation / interpolation techniques. But his list of proscribed gridded products includes many products that later appear in tables and discussion: CDIAC, CEDS, EDGAR, etc. In some cases I know that both national and gridded versions exist, but in other cases I would need to check (meaning that I do not understand offhand the distinction). Other readers may also not know nor understand this initial distinction. Figure 1 shows six primary and five secondary products (total = 11, including gridded products). Of these, the author goes on to provide details of all but ODIAC (one of the gridded products). Table 1 (useful, necessary!) lists nine sources, omitting ODIAC and CAIT. Sections 5.1 through 5.10, however, provide textual detail on 10 sources, now including CAIT. In Figure 3 the reader again encounters eight data products, now excluding ODIAC, CAIT and (surprisingly) UNFCCC CRFs. I assume the author has valid reasons for inclusion or exclusion of specific products in each of various tables, figures and paragraphs, but those reasons escape me. I have few doubts about the magnitude of the effort nor about the skill required, but the organization seems to distract?

Specific comments

Page 2, line 14: "reasons why estimates differ between datasets, but this requires". Please resolve singular / plural: 'reason … requires' or 'reasons … require'.

Page 2, lines 16, 17: "not all datasets attempt to be comprehensive either geographically or by including all emissions sources". True, perhaps, but they all appear under a 'global' label with implicit 'comprehensiveness'? A technical methodological fault clouded by mis-leading self-promotion? Not much the author can do if a data product claims global coverage but misses recent years or ignores key sectors? See note on possible recommendations, above.

Page 3, line 5: "This" change to 'that', referring to Macknick 2011

Page 4, line 1: "In so doing he presents of emissions estimates from the global combustion of coal …" Something wrong or missing in this sentence?

Page 4, lines 3 to 5: "Guy Callendar, investigating the influence of fossil …" This sentence would make more sense and show consistency with the prior sentence if author moved the citation ahead, to just after the name: 'Guy Callendar (Callendar 1938), investigating the …' Same for line 6, Gilbert Plass. By these changes (e.g. moving the citations and therefore the dates to the start of each sentence), readers can better follow the time sequence of these early estimates.

Page 4, line 19: "They" here refers to the advisory panel, to the Revelle et al citation, or to? Please clarify.

Page 5, line 4: "alternative source of energy data" - Alternate to what? What alternate? One of these sources, original or alternative, eventually evolves into the basis for CDIAC?

Page 5, lines 8,9: "… parameters, still constant in time, an assumption …" - the phrase 'constant in time' invites confusion here. You mean parameters such as carbon content of fuels, combustion efficiency, etc. were used by Marland and Rotty as fixed (invariant) through time and across countries? But CDIAC eventually and now used source-specific and country-specific combustion efficiencies? On line 11 you refer to these country-specific factors? You also write (lines 11, 12) about avoidance of the "use of global-average conversion factors". What constant parameters were used in the original paper and what time-variant replacements (improvements) occurred as a consequence of Marland and Rotty (better that you should tell us based on your knowledge rather than that we each should apply our own interpretations)?

Overall comments, Section 2: I enjoyed historical accounting and consider it of immense value. To have it now in one place, close by all the current estimates, seems intensely useful. From my training, however, I missed discussion of the Suess effect related to bomb $^{14}$C? Keeling certainly published on that by 1979 (I had to go back to look) and Hans Suess (cited earlier) must have done likewise. Perhaps not so much related to quantitative emissions but - given many side comments in this section about source assumptions - identification (first) and quantification (later) of the Suess effect pretty much nailed fossil fuels as the causative source of growing CO2 concentrations? Author discusses the (Seuss?) dilution effect with reference to Baxter & Walton 1970 but other authors, including Suess, had it earlier? Perhaps those earlier publications did not follow through to actual quantitative emissions? No specific changes here, only curious about how what I thought I learned fits with the author's re-counting of events.

Page 5, line 28: "flue gas desulphurisation" need a valid reference here.

Page 5, lines 29, 30 - clarify by slight changes in wording and punctuation: "Datasets may exclude carbonate emissions entirely or include emissions only from cement production (e.g., CDIAC) or from all carbonate decomposition (e.g., EDGAR)."

Page 6, line 4: "from land-use change and carbonates combined" - this phrase implies that you want combined emissions from LUC and carbonates. But, in fact, you mean 'from land-use change and from combined carbonate sources'? With this minor change, sentence now fits better with 13% and 5% from most recent GCP budget?

Page 9, line 19: "Eurostat (IEA, no date-a)." Author uses this designation to link to distinct EIA and IEA URLs in the reference list. Copernicus / ESSD perhaps have a preferred format? Typesetters / proofreaders will pick this up?

Page 12, line 23: "because of high non-fuel use in oil" - non-fuel use of oil?

Page 12, line 28: "oil is fully disaggregated" - what does this mean? Oil use already fully disaggregated (e.g. no further detail) or oil fully disaggregated into source types, refined uses, non-combustion uses, etc?

Page 13, line 46 (line numbers turn continuous in Section 5?): "the other two datasets". Which 'other two' datasets? CDIAC and IEA? Acronym confusion here.

Page 13, line 48: any uncertainty data for CAIT?

Page 13, line 52: CEDS = gridded!? As expected if useful for CMIP6?

Page 14, line 94: "number of small countries (see Figure 1)" - Figure 1 shows CDIAC compiled from UNFCCC reports plus independent $CO_3$ data but shows nothing about inclusion or exclusion of small countries?

Page 15, line 128: "emissions as ±10% at 95%/2sd" - at least at first use, write this out as 95% Confidence Interval / 2 standard deviations?

Page 15, line 141: "more complete v5.0 provides" - v5.0 here refers to the v5.0 Crippa et al. source listed in Table 1. That source also publicly available, e.g. in reference to v5.0 FT version referred to a few lines earlier (on line 139). So the FT version has available and non-available versions while the definitive v5.0 always exists in available form? Possible confusion here?

Page 16, lines 152, 154: I realize author needs to adopt uncertainty units as provided by each of the sources but here one encounters "2σ" where earlier (for CDIAC) we had 95% CI and 2 standard deviations. Could the author apply and report uncertainties in a uniform set of units, perhaps with a prefatory note that 95% CI ~= 2 sd = 2σ?

Page 16, line 156: "EIA is a federal statistical agency formed" - write instead a 'US federal statistical agency? Small change here will clarify reference to US data in subsequent paragraphs.

Page 17, lines 204, 205: "GCP reports uncertainties at 68%/1sd level (Friedlingstein et al., 2019)." Because you have, in prior sentence, referred to emissions uncertainties in GCP at (again) 95% CI / 2 standard deviation, here you should clarify that 68% CI, 1 standard deviation applies to the entire (net) global carbon budget (sources and sinks)?

Page 18, line 236: "41 flows." Explain 'flows' in this context? Or, move the flows definition starting from line 239 to here instead?

Page 18, lines 242, 243: "usually released in October/November" - for the prior year? Because of independent specific reports and (presumably) skilled staff efforts, subject to the same year reporting uncertainty described earlier?

Page 19, line 249: "in addition to differing by statistical differences" - Not sure what you mean here? By statistically valid differences? Or by quantitatively useful statistical differences?

Page 19, lines 257, 258: I think this means that, in working with and responding to IEA, national staff develop knowledge and skills that then improve national ability to comply with other national reporting functions, e.g. to UNFCCC?

Page 19 line 267: At the start of this sentence you refer to the BUR in lower case but at the end of the same sentence you refer to them in upper case. Decide one or the other? Note also line 299, BUR capitalized.

Page 21, line 305: HYDE 3.2 (updated to 2015?) appeared in ESSD (https://doi.org/10.5194/essd-9-927-2017), with a much better description of sources and time extent.

Page 21, line 325: Figure 3 shows eight, not six, emissions data products.

Page 22, line 331: Legend refers to seven data sets but table shows eight?

General comment on Section 6: Packed with information but this reader finds text and graphics very useful!

Page 22, line 335: Data easily available and very easy to use. Thanks.

Page 23, Figure 4: 'WLD' refers to 'world'? (Given the title, replace 'world' with 'global'?) Because EDGAR data goes through 2018, this must be EDGAR v5.0 FT or v5.0 standard? Sub-categories of PRIMAP (Hist-TR and Hist-CR) were NOT described in PRIMAP section 5.9?

Page 24, Figure 5: similar questions to those for Figure 4 but convergence factors (e.g. Page 23 lines 344 to 349) very clear. If, taking GCP as reference, emissions in 2014 amounted to 35 Gt, then 1.7 amounts to only 5%. Substantial improvement? Does this deserve more mention / discussion? Note: I find 5% number repeated around line 655 but still without much emphasis? Author should, rightfully, claim or proclaim this number as a not-automatic outcome of sorting through a mass of information? Best case under current limitations?

Page 24, line 363: "in liquid fuels that IEA because" - 'than' instead of 'that'?

Page 25, line 368: "because of statistical differences" - again, what does the author mean by use of the term 'statistical differences'?

Page 25, line 385: "global energy consumption from three energy datasets" - list the three data sources (IEA, BP, EIA), otherwise readers needs to go to Figure 7 to know what the author refers to? Why include EIA here given its earlier outlier status?

Page 26, line 393: "BP's oil consumption numbers lie slightly below those of the IEA and EIA over the entire period". Not true in Figure 7 as presented.

Page 26, line 396: "Gross Calorific Value (GCV; Higher Heating Value)" - Why does reader encounter 'Higher Heating Value' here. Term only used on this page and in earlier usage (line 389) author did not capitalize. Not clear what value the term adds?

Page 39, line 614: "emissions through to over 6000 Gt CO2" - What does the author want to say here? Not clear.

---

## Short Comment (SC1) · 10 May 2020

Fantastic paper, including the extensive Supplementary Information!

Some minor comments you might want to consider or might be relevant for readers of this paper below

**CAIT**

According to https://www.climatewatchdata.org/ghg-emissions emissions until 2016 are available,.

"Description: As of December 2019, CAIT Historical Emission data contains sector-level greenhouse gas (GHG) emissions data for 185 countries and the European Union

(EU) for the period 1990-2016, including emissions of the six major GHGs from most major sources and sinks."

**CEDS**

There has been a new maintenance release (after the cited 2019-12-23 release): https://github.com/JGCRI/CEDS/wiki/Release-Notes

**PRIMAP-hist**

PRIMAP-hist is sometimes written as PRIMAP-Hist in the manuscript.

A reference for the PRIMAP emissions module is Nabel et al. (https://doi.org/10.1016/j.envsoft.2011.08.004).

The data set from the Detailed-Data-By-Party interface has also been updated and is available from Zenodo:

Gieseke, Robert, & Johannes Gütschow. (2020, April 23). UNFCCC Emissions data from the Detailed Data By Party interface (Version 2020-03-30). Zenodo. http://doi.org/10.5281/zenodo.3763020

**CRF**

Line 277: Did they really agree on Excel as a tool or just on the layout of the data which was to be reported?

There has been a recent update to the PRIMAP-crf dataset, which eases access to the UNFCCC's Excel files:

Gütschow, Johannes, Jeffery, M. Louise, & Günther, Annika. (2020). PRIMAP-crf: UNFCCC CRF data in IPCC categories (PRIMAP-crf-2019-v2) (Version 2019v2) [Data set]. Zenodo. http://doi.org/10.5281/zenodo.3775575

**Figures Reading the line graphs is really helped by the matching height of the legend items. Just some colors are hard to see (yellow) and maybe the lines could be made a**

little thicker?

---

## Author Comment (AC1) · 27 May 2020

**Review ESSD-2020-34-RC1**

Comparing global CO2 emissions is no small undertaking, but immensely important in the modern era as space-based measurements need ground verification. Robbie Andrew does a thorough job in stating the problem, describing the different boundary conditions of each dataset, and doing a more intensive comparison of selected countries or groups of countries. This is similar to the efforts of Andres et al. 2012 and Macknick 2011, but there is a need for a newer update. Most of my commentary is for copy edits and some clarification of sources and language/terminology, bit I recommend this for publication with minor revisions.

I thank the reviewer for their careful and helpful review of the article.

Line by line comments
Page 1, line 24-25: Is there a source for this somewhere?
This is actually one of my findings, so in that sense it introduces the material and analysis in the article. It is the subject of section 2, on early estimates. So no, I have no reference for this sentence.
- Changes made: none.

Line 26: "realm of scientific enquiry"
- Changes made: Added word 'of'

Page 2, line 7-8: A simple listing of these core reasons would be helpful to the reader
This is the introductory section, so rather than spelling these out here, I have pointed ahead to later discussion.
- Changes made: Added to end of sentence "; these will be discussed in detail in this article"

Line 9-10: What exactly is the total budget we are drawing near? More detail would be helpful here, especially in connecting with section 7
If the question of what the total budget is had a simple answer, I would certainly address it here. But there are many variants of the value based on different methods used for determining it, different temperatures, and uncertainty that is very difficult to communicate (ensembles of convenience, etc.). There is considerable literature on this subject that is very difficult to summarise concisely here, and providing one or two numbers would be highly misleading. I believe that the reference here to the CONSTRAIN project (one of the most recent efforts) and later in section 7 to Rogelj et al 2016, is sufficient.
- Changes made: none.

Line 11-13: Does this really helps reduce uncertainty, or rather identifies potential sources of uncertainty?
The reviewer is pointing towards the issue of perceived uncertainty vs actual uncertainty, and it's true I haven't been clear here. One might say that uncertainty in global emissions is perceived to be higher than it actually is when one understands the data better, but one could argue that a better understanding of the data and recognition that certain differences are not measurement error, reduces uncertainty.
- Changes made: reworded "important in efforts to reduce uncertainty" to "important in efforts to correctly represent uncertainty"

Line 19-22: a list of acronyms used would be helpful for the reader
- Changes made: Added footnote at the end of the sentence: "CDIAC: Carbon Dioxide Information Analysis Center; ODIAC: Open-Data Inventory for Anthropogenic Carbon dioxide; EDGAR: Emissions Database for Global Atmospheric Research; FFDAS: Fossil Fuel Data

Assimilation System; CEDS: Community Emissions Data System; PKU-FUEL: Peking University Fuel emissions."

Page 3, Line 3-7: Regular updates to procedures like this (Comparisons of FFCO2 from different sources) are important, but maybe a connection to the IPCC's 1.5 C report as to why this is especially relevant to address.

Thank you for this suggestion.

- Changes made: Added sentence "Given the introduction of a temperature-based goal in the Copenhagen Accord (UNFCCC, 2009), and in particular the much more proximal 1.5°C goal in the Paris Agreement (UN, 2015), tracking global emissions is highly important, requiring reliable estimates of uncertainty."

Page 4, Line 28-30: What were Keeling's global and cumulative estimates, and how do they compare to the previous estimates? Don't know if that is available, but would be interesting to add.

I hadn't thought to add much detail in this section of the paper, but for your interest, you can read my earlier, longer discussion here: http://folk.uio.no/roberan/t/EarlyEstimates1.shtml.

- Changes made: none.

Page 5, line 13: "estimates had be based"

The text is currently "that previous estimates had been based on", which is grammatically correct.

- Changes made: None.

Page 5, line 29: (REF)?

Oops!

- Changes made: Added a reference to Córdoba, P., 2015. Status of Flue Gas Desulphurisation (FGD) systems from coal-fired power plants: Overview of the physic-chemical control processes of wet limestone FGDs. Fuel 144, 274-286. DOI: https://doi.org/10.1016/j.fuel.2014.12.065. 0016-2361.

Page 7 line 19-20: Is there a reason for the separation? Or just a discipline versus generic issue?

I don't know for sure why this happened and was unwilling to speculate in the article. Since I've worked in both emissions accounting and economics, and have served my time mapping emissions data to economic sectors, this jumped out at me, and I've been aware for a long time that many have misunderstood this, so wanted to point it out here. IPCC sectors are defined in a way that is suitable for emissions accounting, and my guess is that those who developed the guidelines had less knowledge of economic statistics and simply misused the term 'sectors of the economy'. But really the problem is more basic than just the use of the term 'sectors of the economy', since that's only mentioned once that I could find, and it's the use of the word 'sectors' that has caused the confusion. People when reading that define it according to their own understanding, too often without question. Even in economics the idea of a sector is highly problematic, with each company being assigned to a sector based on its majority activity, and there are many companies having more than one activity. I'll leave the text here as it stands.

- Changes made: none.

Page 14 line 96: Reference needed for USGS minerals yearbook.

- Changes made: Added reference to "USGS: Cement, in: 2016 Minerals Yearbook - Metals and Minerals, 2020. https://www.usgs.gov/centers/nmic/cement-statistics-and-information (Last access: 26 May 2020)."

Page 15 line 128-129: The estimates of uncertainty needs to be on a consistent format (See also lines 152 and 203)

- Changes made: CDIAC is the first dataset I discuss that includes uncertainty information. So there I have now spelled out "confidence interval" and "standard deviations" in the first use and used simply "95%/2σ" in the second instance in the same sentence. Thereafter I follow this second format as appropriate.

Page 24, line 363: "in liquid fuels than"
- Changes made: Corrected "that" to "than".

Page 28, line 442-444: Any inclination as to why there is this difference between the other datasets and IEA/CDIAC? Treatment of nonenergy uses maybe? Carbon content of fuel types?
This is discussed on page 30, from line 462 of the review version.
- Changes made: none.

Page 33, line 32: IEA, not IEwwA
- Changes made: IEwwA corrected to IEA.

Page 34 line 33: "one can see more"
- Changes made: Corrected "one can see the more" to "one can see more".

Page 40 line 637: Would a recommendation to data producers be to make their methodology and data sources more transparent to allow for better understanding?
The difficulty I have here is what concrete suggestions can be made for doing this. I know that different groups have thought a lot about how to transparently and comprehensively document methods and sources, but still we come up with a lot of missing and obscured information. I tend to think this is almost unsolvable. While it might be thought that the best approach might be to have a standard format, the methods differ quite substantially such that it's not necessarily straightforward to report them in the same way.
Instead, I have added a new paragraph that partly addresses the reviewer's comment.
- Changes made: Added new paragraph in Discussion section: "Given the inconsistent system boundaries across emissions datasets, one could conceive of a 'carbon emissions dataset intercomparison project', or CEDIP, along the lines of the Coupled Model Intercomparison Project (CMIP) and other related model comparison projects. A core part of these intercomparison projects is the requirement that participants report model outputs to a specified and very clear template such that the issue of system boundary differences is removed. For example, an estimate for global total $CO_2$ emissions could not be reported for a dataset if it excluded sources or countries. In effect, a CEDIP would extend the work done in this article, allowing each data provider to submit according to their superior understanding of their own datasets, permitting more robust comparison, and, critically, allowing lessons to be gained such that estimates can be improved."

Page 41 line 656: What would a better measure of uncertainty be? This is a good measure of variance in system boundaries and how they relate to outputs though.
Once (if ever) we have reporting on a consistent basis with consistent labelling of 'totals' — as for example via a CEDIP — then we can more reliably use range to indicate uncertainty. But this still doesn't get away from the issue of interdependence of datasets: when the datasets are based on entirely separate efforts and separately collected data, then we have independence, but how is that possible? This approaches the 'ensemble of convenience' issue as well: that we happen to have only these datasets, so our estimate of uncertainty will be biased. I have touched on these issues throughout the article, but there remain a lot of questions and issues here that go a little beyond the scope I had envisaged for the paper.
- Changes made: none.

Page 42 line 690-693: What about space-based measurements?
This is in fact what is meant by "atmospheric inversions" in the previous paragraph, and this wasn't sufficiently clear.

- Changes made: Added sentence "The use of atmospheric inversion modelling, derived from satellite observations but still requiring a priori estimates, is an active area of research."

**Review ESSD-2020-34-RC2**

Substantial timely effort to resolve "range = uncertainty" "tangle". Large challenging effort for which time and expertise required remain hidden or at least understated.

Overall, a remarkable effort of immense value. Everyone involved in emissions and budget calculations, as well as larger numbers of folks engaged in numerous (often divergent) political approaches and strategies around lowering those emissions, should read at least the cautionary Discussion (Section 8). Authors and advocates of various emission products, even if their particular product does not appear here, need to learn from this work. If pressed, I would recommend Figure 3 and Figure 5, but specific countries and specific analysts will need to explore carefully through all details of Section 6. I strongly recommend publication in ESSD!
Many thanks for your encouraging comments! Thank thank you for your careful and helpful review.

Author several times notes the absence of a definitive reference emission data product. Add a statement, at least once, that extends that statement to the inevitable conclusion that no data product discussed here can claim fully-global all-sector coverage despite their titles. (Perhaps GCP comes closest?) Author several times notes benefit of well-staffed data compiling organizations (e.g. IEA) and of positive outcomes enhanced by training and consultation (e.g. in EU). Bring these positives forward into a recommendation, perhaps near the end of Section 8? Recommendations as they stand (mostly sotto voce) seem to imply more work to tease out differences. Call out at least a few good examples that might serve to minimize errors and differences? Set yourself or colleagues up for a good EU proposal to accomplish some of the needed rectification?
In fact, EDGAR comes closer to that goal than GCP does, since GCP does not yet include emissions from carbonates other than cement.
Perhaps I'm a shrinking violet, but I have added one paragraph to the Discussion section in response to the reviewer's call for action.

- Changes made: Added new paragraph in Discussion section: "Given the inconsistent system boundaries across emissions datasets, one could conceive of a 'carbon emissions dataset intercomparison project', or CEDIP, along the lines of the Coupled Model Intercomparison Project (CMIP) and other related model comparison projects. A core part of these intercomparison projects is the requirement that participants report model outputs to a specified and very clear template such that the issue of system boundary differences is removed. For example, an estimate for global total $CO_2$ emissions could not be reported for a dataset if it excluded sources or countries. In effect, a CEDIP would extend the work done in this article, allowing each data provider to submit according to their superior understanding of their own datasets, permitting more robust comparison, and, critically, allowing lessons to be gained such that estimates can be improved."

This reviewer remains a bit confused about which data sets the author evaluates and why. Early in the manuscript, the author disclaims attention to gridded emissions products. For good reason - this manuscript focuses on source ("system boundary") uncertainties rather than disaggregation / interpolation techniques. But his list of proscribed gridded products includes many products that later appear in tables and discussion: CDIAC, CEDS, EDGAR, etc. In some cases I know that both national and gridded versions exist, but in other cases I would need to check (meaning that I do not understand offhand the distinction). Other readers may also not know nor understand this initial distinction. Figure 1 shows six primary and five secondary products (total = 11, including gridded products). Of these, the author goes on to provide details of all but ODIAC (one of the gridded products). Table 1 (useful, necessary!) lists nine sources, omitting ODIAC and CAIT. Sections 5.1 through 5.10, however, provide textual detail on 10 sources, now including CAIT. In Figure 3 the reader again encounters eight data products, now excluding ODIAC, CAIT and (surprisingly) UNFCCC CRFs. I assume the author has valid reasons for inclusion or exclusion of specific products in each of various tables, figures and paragraphs, but those reasons escape me. I have few doubts about the magnitude of the effort nor about the skill required, but the organization seems to distract?

Yes, I haven't been very consistent here, and I was aware of that without having a clear idea of how this can be resolved.

Primarily, as you say, I wanted to focus on datasets that provide national and global emissions estimates, without the requirement to process the dataset first to obtain these (e.g., using country boundaries and GIS methods), since that is beyond the capacity of most data users. So the datasets I describe in detail are those that readily provide estimates of national and global $CO_2$ emissions.

The specific reason I added ODIAC to Figure 1 was to address a confusion I observed in the inversion-modelling community, the details of which now escape me, but it felt necessary to point out how ODIAC was related to other datasets. This has indeed introduced a minor inconsistency in the way I've treated the gridded datasets.

Further, the reason I omitted CAIT from Table 1 and the analysis was simply because it is such a simple combination of other emissions datasets, but the line is perhaps blurred between that and PRIMAP-hist, which also is largely (not entirely) a combination of other emissions datasets. Both are excluded from the comparative analyses in the paper for this reason, which is really a matter of degree rather than being categorical. But given that these two datasets are largely derived from other emissions datasets, I didn't feel that there was much to gain by comparing them to other datasets and then repeatedly observing that they were similar for this very reason.

The UNFCCC CRFs were excluded from Figure 3 simply because these do not include an estimate of global emissions. But I see this was unnecessarily strict.

One reason Table 1 includes PRIMAP-hist because that is used in section 7, on cumulative emissions. The reason PRIMAP-hist is included here is because they are one of few with a long time series. The focus of the paper isn't on cumulative emissions, so this is a short excursion, without having gone to the effort of understanding why these three datasets are so different in a cumulative sense. I have simply made this observation and left this for future work, since I have to draw a line somewhere!

So all in all there are reasons for the choices made in the article, although I agree they can appear a little arbitrary. The question then is how important it is that the reader understand these choices. The reviewer suggests this is a distraction, and perhaps I would think the same thing if I were reviewing the article, but I believe much less so if I were coming to it as a reader.

Some of my rationale for inclusion/exclusion can be readily debated, but in the end I don't believe it is a significant distraction for the reader.

- Changes made: Added UNFCCC CRFs to Figure 3, removed word 'global' from caption.

Specific comments

Page 2, line 14: "reasons why estimates differ between datasets, but this requires". Please resolve singular / plural: 'reason … requires' or 'reasons … require'.

- Changes made: Changed sentence structure, From "There are …, but this requires a close examination … to determine …." to "There are …, but a close examination of … is required to determine …."

Page 2, lines 16, 17: "not all datasets attempt to be comprehensive either geographically or by including all emissions sources". True, perhaps, but they all appear under a 'global' label with implicit 'comprehensiveness'? A technical methodological fault clouded by mis-leading self- promotion? Not much the author can do if a data product claims global coverage but misses recent years or ignores key sectors? See note on possible recommendations, above.

I wouldn't go as far as characterising this as self-promotion. If only those datasets that included every single territory and emission source were published, then we'd have much less to work with. The problem is a general lack of clear documentation and understanding among users, and this article is my effort to address this. I do hope more will be done in addition to what I have managed here.

Page 3, line 5: "This" change to 'that', referring to Macknick 2011

- Changes made: Changed 'this' to 'that', as suggested.

Page 4, line 1: "In so doing he presents of emissions estimates from the global combustion of coal ..." Something wrong or missing in this sentence?

- Changes made: Reworded to "presents estimates of emissions".

Page 4, lines 3 to 5: "Guy Callendar, investigating the influence of fossil ..." This sentence would make more sense and show consistency with the prior sentence if author moved the citation ahead, to just after the name: 'Guy Callendar (Callendar 1938), investigating the ...' Same for line 6, Gilbert Plass. By these changes (e.g. moving the citations and therefore the dates to the start of each sentence), readers can better follow the time sequence of these early estimates.

- Changes made: Shifted the two citations as recommended.

Page 4, line 19: "They" here refers to the advisory panel, to the Revelle et al citation, or to? Please clarify.

- Changes made: Changed "They also broke emissions down" to "The Panel's report also broke emissions down".

Page 5, line 4: "alternative source of energy data" - Alternate to what? What alternate? One of these sources, original or alternative, eventually evolves into the basis for CDIAC?
The point here was simply that he was apparently the first to look at more than one energy data set to address the question of uncertainty, and which other dataset isn't important for making that point. The dataset in question was the US Bureau of Mines' Minerals Yearbook. The way I've worded this has led to ambiguity.

- Changes made: Prepended the following text to the sentence: "While most previous estimates had relied on energy data from the UN".

Page 5, lines 8,9: "... parameters, still constant in time, an assumption ..." - the phrase 'constant in time' invites confusion here. You mean parameters such as carbon content of fuels, combustion efficiency, etc. were used by Marland and Rotty as fixed (invariant) through time and across countries? But CDIAC eventually and now used source-specific and country- specific combustion efficiencies? On line 11 you refer to these country-specific factors? You also write (lines 11, 12) about avoidance of the "use of global-average conversion factors".
What constant parameters were used in the original paper and what time-variant replacements (improvements) occurred as a consequence of Marland and Rotty (better that you should tell us based on your knowledge rather than that we each should apply our own interpretations)?

- Changes made: Reworded "constant in time" to "time-invariant"
- Changes made: Reworded "slightly revised parameters" to "slightly revised emission factors"
- Changes made: Reworded "the use of global-average conversion factors" to "the use of global-average, time-invariant energy conversion factors"

Overall comments, Section 2: I enjoyed historical accounting and consider it of immense value. To have it now in one place, close by all the current estimates, seems intensely useful. From my training, however, I missed discussion of the Suess effect related to bomb $^{14}$C? Keeling certainly published on that by 1979 (I had to go back to look) and Hans Suess (cited earlier) must have done likewise. Perhaps not so much related to quantitative emissions but - given many side comments in this section about source assumptions - identification (first) and quantification (later) of the Suess effect pretty much nailed fossil fuels as the causative source of growing CO2 concentrations? Author discusses the (Seuss?) dilution effect with reference to Baxter & Walton 1970 but other authors, including Suess, had it earlier? Perhaps those earlier publications did not follow through to actual quantitative emissions? No specific changes here, only curious about how what I thought I learned fits with the author's re-counting of events.
I can't say I've read closely about the Suess effect, and I only mentioned it briefly as the motivation of Baxter & Walton. When researching this part, I was largely focussed on how emissions estimates had been derived, what their data sources were, what assumptions they'd made, and so on. Sorry I can't help further!

- Changes made: none.

Page 5, line 28: "flue gas desulphurisation" need a valid reference here.

- Changes made: Added a reference to Córdoba, P., 2015. Status of Flue Gas Desulphurisation (FGD) systems from coal-fired power plants: Overview of the physic-chemical control processes of wet limestone FGDs. Fuel 144, 274-286. DOI: https://doi.org/10.1016/j.fuel.2014.12.065. 0016-2361.

Page 5, lines 29, 30 - clarify by slight changes in wording and punctuation: "Datasets may exclude carbonate emissions entirely or include emissions only from cement production (e.g., CDIAC) or from all carbonate decomposition (e.g., EDGAR)."

- Changes made: As suggested.

Page 6, line 4: "from land-use change and carbonates combined" - this phrase implies that you want combined emissions from LUC and carbonates. But, in fact, you mean 'from land-use change and from combined carbonate sources'? With this minor change, sentence now fits better with 13% and 5% from most recent GCP budget?

- Changes made: Removed word 'combined.'

Page 9, line 19: "Eurostat (IEA, no date-a)." Author uses this designation to link to distinct EIA and IEA URLs in the reference list. Copernicus / ESSD perhaps have a preferred format? Typesetters / proofreaders will pick this up?
From experience I know that editing in ESSD is quite thorough, so I'll let them sort this out if/when we come that far.

- Changes made: none.

Page 12, line 23: "because of high non-fuel use in oil" - non-fuel use of oil?
Yes, that's certainly better. I had been thinking of 'oil' as a category of emissions rather than a substance at this point, and shouldn't expect readers to follow that thinking!

- Changes made: Changed from "in" to "of".

Page 12, line 28: "oil is fully disaggregated" - what does this mean? Oil use already fully disaggregated (e.g. no further detail) or oil fully disaggregated into source types, refined uses, non-combustion uses, etc?

- Changes made: Capitalised 'oil' to be consistent with 'coal' and 'gas' later, and changed "while oil is fully disaggregated" to "while Oil is divided into a number of petroleum products"

Page 13, line 46 (line numbers turn continuous in Section 5?): "the other two datasets". Which ' other two' datasets? CDIAC and IEA? Acronym confusion here.
Yes, I originally had the sections in a different order, so that these acronyms had already been defined, and neglected to fix this after re-ordering.

- Changes made: Changed "the other two datasets" to "either IEA or CDIAC data".
- Changes made: Changed to "For CO2, IEA's (International Energy Agency)…" and "EIA (US Energy Information Administration) data…".

Page 13, line 48: any uncertainty data for CAIT?

- Changes made: Added "The WRI refers readers to the underlying data sources for information on uncertainty, and makes no assessment of uncertainty in their assembled dataset."

Page 13, line 52: CEDS = gridded!? As expected if useful for CMIP6?

- Changes made: Added sentence: "In addition to gridded estimates, CEDS also produces country-level estimates, which are those discussed here."

Page 14, line 94: "number of small countries (see Figure 1)" - Figure 1 shows CDIAC compiled from UNFCCC reports plus independent $CO_3$ data but shows nothing about inclusion or exclusion of small countries?

CDIAC doesn't use data from UNFCCC, but rather from the UN Statistics Division, which is shown in the yellow energy data column. These data include small countries. UNFCCC is shown separately in the figure in the blue primary emissions data column.

- Changes made: Added a sentence to the caption of Figure 1: ""UN stats" is the United Nations Statistics Office (not UNFCCC)."

Page 15, line 128: "emissions as ±10% at 95%/2sd" - at least at first use, write this out as 95% Confidence Interval / 2 standard deviations?

- Changes made: Written out, as suggested.

Page 15, line 141: "more complete v5.0 provides" - v5.0 here refers to the v5.0 Crippa et al. source listed in Table 1. That source also publicly available, e.g. in reference to v5.0 FT version referred to a few lines earlier (on line 139). So the FT version has available and non-available versions while the definitive v5.0 always exists in available form? Possible confusion here?

Yes, there's certainly possible confusion. The problem here is that I have been unable myself to obtain clarity on this question. If there remains confusion, then I'm reflecting the situation at the data provider. Data versioning and availability could be made clearer by the data provider, but I'm reluctant to make such a bald statement in the article.

- Changes made: none.

Page 16, lines 152, 154: I realize author needs to adopt uncertainty units as provided by each of the sources but here one encounters "2σ" where earlier (for CDIAC) we had 95% CI and 2 standard deviations. Could the author apply and report uncertainties in a uniform set of units, perhaps with a prefatory note that 95% CI ~= 2 sd = 2σ?

- Changes made: CDIAC is the first dataset I discuss that includes uncertainty information. So there I have now spelled out "confidence interval" and "standard deviations" in the first use and used simply "95%/2σ" in the second instance in the same sentence. Thereafter I follow this second format as appropriate.

Page 16, line 156: "EIA is a federal statistical agency formed" - write instead a 'US federal statistical agency? Small change here will clarify reference to US data in subsequent paragraphs.

- Changes made: Added the word "US" as suggested.

Page 17, lines 204, 205: "GCP reports uncertainties at 68%/1sd level (Friedlingstein et al., 2019)." Because you have, in prior sentence, referred to emissions uncertainties in GCP at (again) 95% CI / 2 standard deviation, here you should clarify that 68% CI, 1 standard deviation applies to the entire (net) global carbon budget (sources and sinks)?

Yes. I mention this because I have seen that some people have not picked this up.

- Changes made: Changed to "GCP reports uncertainties for all components of the global carbon budget at 68%/1sd level."

Page 18, line 236: "41 flows." Explain 'flows' in this context? Or, move the flows definition starting from line 239 to here instead?

- Changes made: Added sentence: "The IEA uses the term flow to describe what happens to energy products, largely categories of production, transformation, or use. Flows are presented at different levels of detail in different datasets."

Page 18, lines 242, 243: "usually released in October/November" - for the prior year? Because of independent specific reports and (presumably) skilled staff efforts, subject to the same year reporting uncertainty described earlier?

I see I have been inconsistent here. The publication time and reporting lag are things that can quickly change, so I had not intended to mention this for all datasets.

I'm afraid I don't understand the reviewer's second question.

Page 19, line 249: "in addition to differing by statistical differences" - Not sure what you mean here? By statistically valid differences? Or by quantitatively useful statistical differences?

- Changes made: Put statistical differences in quotes, and changed parenthetical text to "which represent the mismatch between supply-side and demand-side data".

Page 19, lines 257, 258: I think this means that, in working with and responding to IEA, national staff develop knowledge and skills that then improve national ability to comply with other national reporting functions, e.g. to UNFCCC?

Partly. What I was thinking was that when the EIA or BP access national energy reports from countries that IEA has been actively involved with, the reports are more consistent, detailed, and with fewer errors. So it's particularly these one-way lines of communication I was thinking of, where organisations (or individuals) simply read the reports, without the two-way interaction.

- Changes made: Reworded sentence as "The process of data harmonisation and training actively driven by the IEA with its close interaction with national agencies helps to develop skills and processes, directly benefitting other organisations that collect directly from these national agencies."

Page 19 line 267: At the start of this sentence you refer to the BUR in lower case but at the end of the same sentence you refer to them in upper case. Decide one or the other? Note also line 299, BUR capitalized.

Ah. The editors of a previous paper removed my capitalisation of 'biennial update report', which I never understood, and the lesson hasn't sunk in, so I've been inconsistent here!

- Changes made: All title-case occurrences changes to lower case.

Page 21, line 305: HYDE 3.2 (updated to 2015?) appeared in ESSD (https://doi.org/10.5194/ essd-9-927-2017), with a much better description of sources and time extent.

The paper the reviewer points to is indeed an update of HYDE, but does not include estimates of $CO_2$ emissions. I have confirmed directly with the first author, such that this statement already stands in the article: "the $CO_2$ data are no longer updated (pers. comm. Kees Klein Goldewijk, February 2019)." I think this is sufficiently clear and will make no further changes here.

- Changes made: None.

Page 21, line 325: Figure 3 shows eight, not six, emissions data products.

Indeed, I added a couple and then neglected to update here. And now in review I've added one more.

- Changes made: Changed "six" to "nine".

Page 22, line 331: Legend refers to seven data sets but table shows eight?

Same problem as above!

- Changes made: Changed "seven" to "nine".

General comment on Section 6: Packed with information but this reader finds text and graphics very useful!

Great!

Page 22, line 335: Data easily available and very easy to use. Thanks.

Excellent.

Page 23, Figure 4: 'WLD' refers to 'world'? (Given the title, replace 'world' with 'global'?) Because EDGAR data goes through 2018, this must be EDGAR v5.0 FT or v5.0 standard? Sub- categories of PRIMAP (Hist-TR and Hist-CR) were NOT described in PRIMAP section 5.9?

I suspect I will be asked to remove these titles in the figures during later stages of this process, but if not then I'll be sure to replace this as suggested.
- Changes made: none.

I neglected to make the sources clear here.
- Changes made: Appended to caption "Sources: see table 1"

I was sure I had described the two PRIMAP variants, but I see I have not; thank you for picking that up.
- Changes made: Added sentence to section 5.9: "PRIMAP-hist includes two variants, which they call 'scenarios': 'HISTCR' is assembled by giving country-reported data (e.g. CRF, BUR) priority over data from third parties (e.g. CDIAC, FAO), while 'HISTTP' is the reverse."

Page 24, Figure 5: similar questions to those for Figure 4 but convergence factors (e.g. Page 23 lines 344 to 349) very clear. If, taking GCP as reference, emissions in 2014 amounted to 35 Gt, then 1.7 amounts to only 5%. Substantial improvement? Does this deserve more mention / discussion? Note: I find 5% number repeated around line 655 but still without much emphasis? Author should, rightfully, claim or proclaim this number as a not-automatic outcome of sorting through a mass of information? Best case under current limitations?

The question here is about the perceived uncertainty determined by looking solely at the range of estimates from different sources, and the uncertainty assessed to propagate from the underlying activity data and methods. I have not reduced that latter uncertainty in this analysis, only the former, which was never true uncertainty in the first place. A major purpose of the article is precisely to demonstrate this. The 5% (1.7/35Gt) is not a good estimate of uncertainty in global emissions, but demonstrates that when datasets are treated carefully, they are in fairly good agreement. I believe I've made this point reasonably clearly, including stating it in the abstract.
- Changes made: none.

Page 24, line 363: "in liquid fuels that IEA because" - 'than' instead of 'that'?
- Changes made: as suggested.

Page 25, line 368: "because of statistical differences" - again, what does the author mean by use of the term 'statistical differences'?
My first use of this term is on page 19, where I have made some changes to clarify this, as per an earlier comment.
- Changes made: nothing further.

Page 25, line 385: "global energy consumption from three energy datasets" - list the three data sources (IEA, BP, EIA), otherwise readers needs to go to Figure 7 to know what the author refers to? Why include EIA here given its earlier outlier status?
The narrative flow here is to uncover that there is a divergence between EIA and other datasets, and then to explore why this is that case in section 6.1.1. So section 6.1.1 concludes that there is an issue with EIA's estimates of China's energy/emissions, which is explored more fully in the SI.
- Changes made: Added parentheses as suggested.

Page 26, line 393: "BP's oil consumption numbers lie slightly below those of the IEA and EIA over the entire period". Not true in Figure 7 as presented.
Well spotted. I suspect that conclusion was based on an earlier version of the chart with fewer years shown.
- Changes made: Changed "over the entire period" to "over recent years."

Page 26, line 396: "Gross Calorific Value (GCV; Higher Heating Value)" - Why does reader encounter 'Higher Heating Value' here. Term only used on this page and in earlier usage (line 389) author did not capitalize. Not clear what value the term adds?

- Changes made: Removed "Higher Heating Value" from the caption on page 26.

Page 39, line 614: "emissions through to over 6000 Gt CO2" - What does the author want to say here? Not clear.

I agree this wasn't entirely clear. From 0 through >6000 Gt CO2, the relationship is approximately linear.

- Changes made: Reworded to "It has been shown that for cumulative global emissions up to 6000 Gt $CO_2$, global temperature rise can be expected to be approximately linearly related to cumulative emissions, …".

**ESSD-2020-34-SC1**

Fantastic paper, including the extensive Supplementary Information!
Thanks!

Some minor comments you might want to consider or might be relevant for readers of this paper below

**CAIT**
According to https://www.climatewatchdata.org/ghg-emissions emissions until 2016 are available,.
"Description: As of December 2019, CAIT Historical Emission data contains sector-level greenhouse gas (GHG) emissions data for 185 countries and the European Union (EU) for the period 1990-2016, including emissions of the six major GHGs from most major sources and sinks."
Thanks for this.
- Changes made: Added a sentence: "A new edition released in December 2019 extended the time series to 2016 using the same methodology."

**CEDS**
There has been a new maintenance release (after the cited 2019-12-23 release):
https://github.com/JGCRI/CEDS/wiki/Release-Notes
This new release doesn't change the emissions estimates, and the data files from the 2019-12-23 release are still the current ones. I'm aware that a new release is imminent, but it has been for several weeks now, so probably won't make it into the revised version of this article. A line in the sand has to be drawn somewhere!
- Changes made: none.

**PRIMAP-hist**
PRIMAP-hist is sometimes written as PRIMAP-Hist in the manuscript.
- Changes made: Changed all occurrences to consistent capitalization as PRIMAP-hist.

A reference for the PRIMAP emissions module is Nabel et al.
(https://doi.org/10.1016/j.envsoft.2011.08.004).
Interesting, thank you. I do feel that referring to the documentation for the software used for managing the database is going a little far in this paper, and it's something I haven't done for other datasets, so I won't change the text here.
- Changes made: none.

The data set from the Detailed-Data-By-Party interface has also been updated and is available from Zenodo:
Gieseke, Robert, & Johannes Gütschow. (2020, April 23). UNFCCC Emissions data from the Detailed Data By Party interface (Version 2020-03-30). Zenodo. http://doi.org/10.5281/zenodo.3763020
Thanks for this. I've re-arranged the sentence a bit and added this citation.
- Changes made: Sentence on page 20 rewritten as "However, Jeffery et al. (2018) have produced a flat-record format dataset from the Excel files, and data available through the UNFCCC's online 'detailed data by party' interface (UNFCCC, no date) have been re-packaged in machine-readable format by Gieseke and Gütschow (2020), facilitating more widespread analysis."

**CRF**
Line 277: Did they really agree on Excel as a tool or just on the layout of the data which was to be reported?

Thank you for picking this up. Indeed it isn't noted in the report from the meeting I cited that Excel would be used, but rather that the common reporting format should "be formally submitted in both electronic form and hard copy", and the UNFCCC Secretariat then developed the Excel tool which has subsequently been used by all Annex-1 parties.

- Changes made: Sentence rewritten as "At the fifth COP in 1999, it was agreed that Annex I parties would report quantitative inventories in both electronic form and in hard copy via a detailed Common Reporting Format (CRF), due 15 April every year (UNFCCC, 2000), and an Excel tool for this purpose was made available by the UNFCCC Secretariat in January 2000 (Temertekov et al., 2003)."

There has been a recent update to the PRIMAP-crf dataset, which eases access to the UNFCCC's Excel files:
Gütschow, Johannes, Jeffery, M. Louise, & Günther, Annika. (2020). PRIMAP-crf: UNFCCC CRF data in IPCC categories (PRIMAP-crf-2019-v2) (Version 2019v2) [Data set]. Zenodo.
http://doi.org/10.5281/zenodo.3775575
Thank you for this.

- Changes made: Added text "updated by Gütschow et al. (2020)"

**Figures Reading the line graphs is really helped by the matching height of the legend items. Just some colors are hard to see (yellow) and maybe the lines could be made a little thicker?**
Thank you for this comment. If/when it comes to post-acceptance stage, I will generate new versions of the figures for publication and have made a note of your comment.